# Can LLMs Rank Candidates with Missing Sensitive Attributes Fairly?

**Oluseun Olulana**[1]                                                    *omolulana@wpi.edu*

**Fabricio Murai**[1,2]                                                   *fmurai@wpi.edu*

**Elke Rundensteiner**[1,2]                                               *rundenst@wpi.edu*

[1] *Data Science Program, Worcester Polytechnic Institute, Worcester, MA, USA*
[2] *Department of Computer Science, Worcester Polytechnic Institute, Worcester, MA, USA*

**Reviewed on OpenReview:** *https://openreview.net/forum?id=VrAs5EJ11G*

## Abstract

Large language models (LLMs) are increasingly deployed in high-stakes ranking systems used for hiring, lending, and scholarship allocation, raising concerns about fairness, accountability, and ethical use. These challenges are exacerbated in ranking settings where sensitive demographic attributes are unavailable due to legal, ethical, or practical constraints. Inferring such attributes may introduce harm by violating consent requirements, misrepresenting individuals, and reinforcing structural inequities. This work thus investigates the timely question: How is LLM-based fair re-ranking impacted when demographic information is missing? In this context, we study three alternate strategies that span alternate places within the pipeline where demographic inference may be deployed: (1) inferring sensitive attributes using traditional third-party services prior to ranking, (2) directly prompting LLMs to produce fair rankings without explicit mention of attribute inference, and (3) employing a chain-of-thought approach in which LLMs are first prompted to infer attributes and thereafter to perform fairness-aware re-ranking. We compare these strategies across multiple datasets using established group-fairness metrics for ranking. Across the datasets we evaluate, LLMs achieve demographic inference accuracy comparable to leading third-party services. We further observe that LLMs can produce rankings that improve group-fairness metrics without explicitly inferring sensitive attributes, suggesting a possible design space for fairness interventions that avoids direct demographic labeling. In contrast to zero-shot in-context learning, few-shot prompting improves LLM's ability in balancing fairness and utility in our experiments. We conclude by discussing ethical and governance implications of deploying LLMs for fairness-critical ranking tasks. While LLMs offer flexibility under demographic uncertainty, their capacity for implicit inference also raises significant risks if adopted without transparency, evaluation, and institutional oversight. *To support reproducibility and continued research exploration by others, we release our source code and experimental artifacts*[1].

## 1 Introduction

**Background.** In recent times, AI tools have been increasingly utilized in the context of decision-making processes, becoming integral to a wide range of critical applications—from scholarship and grant decisions to job hiring—that profoundly affect humans (Kim et al. (2024)). Such applications involve the allocation of resources in a ranked order, where the position implies relevance or priority (Qin et al. (2024)). As with

---

[1] https://github.com/sewen007/Hidden-FaiReRTabLLM

any transformative technology impacting society, these AI-driven advances have sparked scrutiny regarding potential risks with algorithmic fairness and bias (Elliott (2024)). Biases often emerge when decisions (e.g., ranking candidates for a job position) correlate with sensitive attributes such as sex or race.

While traditional AI-algorithms for incorporating fairness in rankings have been well explored (Zehlike and Castillo (2020)), the emergence of large language models (LLMs) demonstrating ever-increasing capabilities raises the question of whether, and if so, how LLMs could be applied for fair ranking (Sun et al. (2023a;b)). This issue warrants particular attention, as organizations are increasingly seeking to capitalize on recent technological advances such as LLMs across their business tasks in pursuit of perceived productivity gains. Hence, examining the appropriateness, effectiveness, and potential harms is crucial.

A central obstacle for empirical investigations into fairness for ranking-based decision-making is that sensitive demographic attributes (e.g., sex, race, or age) often are not available yet are needed for incorporating fairness constraints Singh et al. (2021). In practice, such attributes may be missing for multiple reasons: legal restrictions on collecting sensitive information, individuals' rights to safeguard their privacy, or practical issues such as incomplete or noisy data (Council of the European Union (2016)). This creates a difficult tension: fairness-aware algorithms generally require access to sensitive demographic data to measure and mitigate disparities, yet these very attributes are often unavailable in real-world systems. We do not advocate demographic inference as a deployment strategy. Instead, we audit how LLM-based ranking pipelines behave under demographic uncertainty, including when demographic signals are inferred, implicit, or absent, and we surface the resulting fairness and governance risks.

**State-of-Art and Their Limitations.** To address the lack of demographic data, prior work has studied the inference of these sensitive attributes using custom inference systems and the resulting impact on ranking fairness (Ghosh et al. (2021; 2023); Olulana et al. (2024)). This line of work highlights a shared element among most pipelines: attribute inference serves as the first step, enabling re-ranking methods to adjust exposure or relevance in a way that mitigates group-level disparities. Nonetheless, reliance on inference introduces new sources of bias and error, raising questions as to whether LLMs can perform this combined task of demographic attribute inference and fairness-aware re-ranking as effectively than traditional systems, or if they can be leveraged for fair re-ranking while bypassing explicit inference. Although LLMs have been used for re-ranking tasks (Sun et al. (2023b)), there is limited work that explores fairness in this context. Gao et al. (2025) consider fairness-aware reranking in special-case settings where sensitive attributes are assumed to be known and correct, and additional user information is available to the reranking model. This motivates the following research questions in the more practical context:

**RQ1 (External inference):** When sensitive demographic data are unavailable, should traditional third-party demographic inference services remain the default for recovering these attributes before LLM-based re-ranking?

**RQ2 (LLM inference):** Could LLMs themselves be leveraged to directly infer sensitive attributes, thereby reducing dependence on external inference services in fairness-aware LLM-based re-ranking frameworks?

**RQ3 (Pipeline design):** When using LLMs for fairness-aware re-ranking under demographic uncertainty, should attribute inference be treated as a separate preprocessing step, or integrated with ranking in an end-to-end approach, and if so, should inference be implicit or guided through chain-of-thought reasoning? These choices define the three prevalent approaches towards pipeline design evaluated in this work.

**RQ4 (Role of demonstrations):** How does the inclusion of few-shot exemplars influence the effectiveness of fairness-aware LLM-based re-ranking across different pipeline designs, LLM models, and datasets?

**Our Approach.** To explore these open research questions, we explore the role of LLMs in promoting fairness within ranking systems when ground-truth demographic data are unavailable. We assume the standard setting where the input list has items or candidates (e.g., scholarship applicants) pre-sorted by some utility criterion (e.g., relevance or performance on the task at hand). In this work, we investigate how various design choices for deploying LLMs impact both fairness and utility outcomes, specifically either by leveraging external demographic signals, by providing only implicit guidance, or by directing the model

through explicit reasoning. These strategies correspond to distinct points along a single design axis, namely, where demographic inference occurs in the pipeline (external, implicit, or integrated), allowing us to study which pipeline stage may be driving fairness outcomes. We intentionally employ a fixed prompt structure across experiments, without task-specific prompt optimization. Fixing the prompts isolates the effect of the pipeline design itself, which is the core theme of our study. In contrast, prompt optimization would confound the measurement, making it more difficult to distinguish whether fairness gains arise from the LLM's underlying capability or from scaffolding we introduced.

**Scope of design choices considered.** We focus on prompting-based interventions (zero-shot, few-shot, and chain-of-thought) rather than more complex expensive alternatives such as fine-tuning, fairness alignment via reinforcement learning from human feedback (RLHF). This scope reflects practical deployment realities: organizations integrating LLMs into ranking pipelines typically rely on hosted models accessed through APIs, where weight-level modifications are unavailable, costly, or prohibited by service terms. Prompting-based approaches are also the most reproducible and auditable, both important properties in fairness-critical settings. We view fine-tuning-based fairness interventions as complementary and a potential direction for future work, particularly in settings where model weights are accessible.

While recent works explored the use of LLMs for the demographic attribute inference subtask, this is the first study to comprehensively evaluate LLM-based strategies for fairness-aware ranking when demographic information is missing.

Our contributions can be summarized as follows:

- **Novel problem framing.** We introduce and formalize the task of LLM-based fairness-aware re-ranking under demographic uncertainty, a research question largely unexplored in the literature. We define a design space organized around where demographic inference occurs (external, implicit, or integrated) and systematically evaluate the core pipeline design choices within this space.
- **Interaction with external demographic inference services.** We analyze how fairness outcomes change when LLM-based ranking methods rely on demographic inference from third-party services. Our results show that fairness-aware re-ranking is highly sensitive to inference quality: low-accuracy services can substantially degrade fairness outcomes, while high-accuracy services yield improvements only under favorable conditions.
- **Auditing LLM demographic inference capacity.** We explore if LLMs themselves can infer sensitive demographic attributes; and show that LLMs can perform demographic inference comparably to third-party services, underscoring that sensitive attributes may be recoverable even when intentionally withheld.
- **Fairness without explicit inference.** We find that certain LLM-based re-ranking strategies can improve group outcomes even without explicit access to sensitive attributes, in some cases approaching the performance of methods that rely on ground-truth demographic data. This suggests a design space for fairness interventions that avoids direct demographic labeling, while raising new questions about transparency and governance.
- **Role of few-shot alignment.** We show that carefully constructed ranking examples empower LLMs to adapt their behavior to align with specific fairness goals, while concurrently improving robustness against errors in demographic inference.

**What this work is NOT tackling.** Our goal is not to position LLMs as replacements for dedicated fair-ranking algorithms, nor to claim that they outperform purpose-built methods when sensitive attributes are available and usable. Instead, we examine the promises and perils of LLM-based re-ranking in real-world settings where sensitive information may be unavailable and its potential impact on fairness toward individuals in decision-making contexts. This focus is motivated by the fact that LLMs are already being deployed in decision pipelines where classical fairness mechanisms are often bypassed or infeasible to apply.

## 2 Related Work

**Fair Ranking Approaches.** Fairness has been incorporated into traditional ranking pipelines through two primary approaches: by introducing fairness constraints directly into the learning objective of the ranking

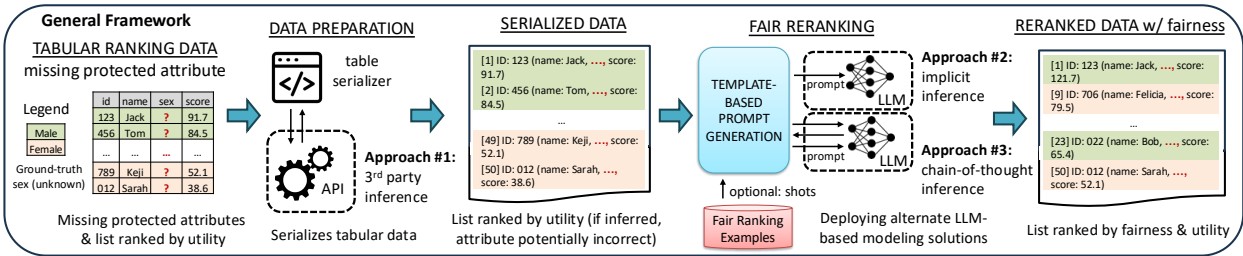

Figure 1: Proposed Framework for LLM-Based Fair Reranking with Sensitive Attributes.

model (Zehlike et al. (2022a;b)), or by post-processing the model's ranked output list through re-ranking (Geyik et al. (2019); Singh and Joachims (2018)). However, despite the growing adoption of LLMs for decision-making tasks and the increasing likelihood that companies may leverage them for fairness-aware ranking applications, this area remains largely underexplored.

**LLMs and Fairness Assessment.** Most work on fairness in LLM-based ranking has centered on evaluating the fairness of model outputs, rather than harnessing LLMs as proactive agents for promoting fairness in downstream ranking tasks (Wang et al. (2024); Liu et al. (2025); Xu et al. (2023)). Other studies evaluating fairness in LLMs include investigations into hiring biases specifically, such as discrimination against candidates with disabilities observed in ChatGPT (Glazko et al. (2024)), demonstration of the amplification of stereotypical biases (Kotek et al. (2023)) or socioeconomic biases (Arzaghi et al. (2024)) by LLMs.

**Fairness with Missing Demographic Information.** Research on fair ranking presumes access to sensitive demographic information to audit or enforce fairness constraints (Zehlike and Castillo (2020); Kotek et al. (2023); Glazko et al. (2024)), including in fairness evaluations involving LLMs (Wang et al. (2024)). Yet, in real-world scenarios, such information may be unavailable due to privacy concerns, legal restrictions, or incomplete data, raising the need for the design of new strategies that do not rely on the availability of sensitive attributes (European Union (2000); U.S. Equal Employment Opportunity Commission (EEOC) (2025)). Research on fairness-aware ranking with uncertainty in demographic attributes is in its infancy. Olulana et al. (2024), and Ghosh et al. (2021) suggest using only highly accurate traditional inference methods; however, both works focused on traditional, non-LLM methods. To our knowledge, no work has has addressed *the research question of uncertain demographic attributes in the context of LLM-based rankings*—the topic of our current work.

**Demographic Inference Using LLMs.** Prior work has studied demographic inference by LLMs as a standalone task or as a tool for bias analysis (AlNuaimi et al. (2024)). For example, demographic attributes are inferred to analyze sexism or persona construction in LLM outputs (Magnossão de Paula et al. (2025)), or are perturbed to examine how non-clinical demographic cues influence downstream decisions such as clinical reasoning (Gourabathina et al. (2025)). A recent survey systematizes fairness approaches under incomplete demographic information across machine learning settings (Wang et al. (2025)), but does not address ranking-specific fairness nor the role of LLMs in re-ranking under demographic uncertainty. In contrast, our work focuses on fair re-ranking when sensitive attributes are unavailable and empirically studies how different LLM-based pipeline designs affect fairness and utility.

## 3 Preliminaries

### 3.1 Fairness-Related Concepts

A sensitive attribute is an individual's characteristic, such as race, sex, gender or age, that is commonly treated as protected under legal and regulatory frameworks and raises heightened concerns around fairness, privacy and consent (U.S. Equal Employment Opportunity Commission (EEOC) (2025); European Union (2000)). All candidates whose attribute values correspond to the same demographic value $a_i$ fall into the

same group, $g_k$. Each candidate $i$ is also associated with a performance score $s_i \in \mathbb{R}$, representing its relevance with respect to the ranking task.

Fairness has been defined in various ways across the literature depending on the context and application (Pitoura et al. (2022)). In ranking scenarios, common fairness objectives include discounted cumulative fairness (Yang and Stoyanovich (2017)), fairness of exposure (Singh and Joachims (2018)), and equity of attention (Biega et al. (2018)). In this work, we focus on group fairness, ensuring that individuals belonging to a certain group are treated fairly relative to those belonging to another group (Zehlike and Castillo (2020)). In our experimental study, we measure the fairness achieved in a ranking with group fairness evaluation metrics such as fairness of exposure and discounted cumulative fairness. See the definition of these metrics in Appendix 5.1.

## 3.2 The Fair Re-Ranking Task

A ranking model can be trained to jointly optimize utility and fairness objectives (Zehlike and Castillo (2020)). Here, *utility* refers to the quality of the ranking with respect to observed relevance scores, typically measured using metrics such as NDCG. *Fairness objectives* refer to constraints or criteria that regulate how exposure or representation is distributed across groups defined by the sensitive attribute. Alternatively, when a utility-optimized model is already available, fairness can be incorporated post hoc by applying a fair re-ranking method to its ranked output, without modifying the underlying model (Zehlike et al. (2017); Geyik et al. (2019)).

**Problem Definition.** Let a ranked list of $n$ candidates be denoted as $C = \{(x_i, s_i)\}_{i=1}^n$, where $x_i$ represents the feature vector of candidate $i$ and $s_i$ its utility or relevance score. The initial ranking is: $\tau_0 = \text{argsort}_\downarrow(s_i)$, that is, the permutation induced by sorting candidates in descending order of utility. Each candidate $i$ is additionally associated with a demographic attribute $a_i \in \mathcal{A}$ with an unknown value. The objective of fair re-ranking is to produce a modified ranking $\tau$ over the same candidate set $C$ that balances utility and fairness with respect to the demographic attribute $a_i$, according to fairness criterion $F$ (Geyik et al. (2019); Zehlike et al. (2017); Singh and Joachims (2018)).

As foundation, fair re-ranking can be formalized as an optimization problem (Singh and Joachims (2018)), where the optimal permutation, $\tau^*$ is defined as

$$\tau^* = \arg \max_{\tau \in \mathcal{S}_n} U(\tau; s) \quad \text{s.t.} \quad \mathcal{L}_{\text{fair}}(\tau; a) \leq \epsilon \tag{1}$$

where $\mathcal{S}_n$ denotes the set of all permutations over $n$ candidates, $U(\tau; s)$ is a utility metric (e.g., NDCG) computed with respect to the score vector $s = (s_1, \ldots, s_n)$, $\mathcal{L}_{\text{fair}}(\tau; a)$ quantifies fairness loss (e.g., exposure disparity or NDKL), and $\epsilon \geq 0$ is a user-defined threshold controlling the acceptable level of unfairness.

In practice, $a_i$ may be missing, making group membership uncertain. Prior work has addressed this by inferring sensitive attributes via third-party services and subsequently applying traditional fair re-ranking methods to the augmented data (Ghosh et al. (2021)), introducing additional assumptions through the inference step.

While Equation 1 specifies a formal fairness criterion, the prompting strategies evaluated in this work instruct the LLM using a general fairness signal rather than an instantiation of the precise formula $\mathcal{L}_{\text{fair}}$. The relationship between this 'underspecified' prompt and the formal objective is deliberate in that it allows us to observe the inherent notion of fairness baked into LLM systems, as we further discuss in Section 7.

## 3.3 Missing Demographic Attributes and Inference

Demographic inference services use other candidate characteristics, like names, email addresses, or photos, to infer demographic information like a person's gender, age, or race. With several such services available, for our experiments, we select three popular methods for gender inference and two for race inference.

For gender inference, we use: (1) **Behind The Name (BTN)**[2] utilizes the etymology and historical usage of names, allowing users to input either a first name or a full name, with data drawn from national statistics

---

[2]https://https://www.behindthename.com/

agencies across various regions. (2) **Gender API (GAPI)**[3] also supports name-based gender inference using the first name or full name. It can enhance accuracy with additional inputs such as email address, location (country, IP address, browser), and publicly available governmental and social media data. (3) **Namsor (NMS)** [4], on the other hand, relies on both the first and last name and boasts extensive global coverage across languages, scripts, and regions. Its model is trained on a diverse dataset of 1.3 million names compiled from baby name statistics, reflecting a wide range of countries, languages, and ethnic backgrounds. For race inference we reuse two race/ethnicity inference setups from the pipeline proposed by Ghosh et al. (2021) to infer those sensitive attributes from full names. (1) **EthCNN** is a character-level convolutional neural network proposed by Kim (2014). (2) **Ethnicolr** is a publicly available library by Chintalapati et al. (2018) (although inspired by Hofstra et al. (2020) ), which provides pretrained name-based race/ethnicity classifiers. These models use neural architectures to capture the relationship between character sequences in personal names and racial/ethnic categories. Table 3 shows the inference accuracies for each dataset used in our experiments against the external inference services (ETCNN and Etnicolr).

**Gender vs. Sex.** Henceforth we opt to use the term 'sex' instead of 'gender' for the reasons explained in Appendix A.4.

## 4 Proposed Framework: LLM-Based Fair Reranking w/ Missing Sensitive Attributes

In this section, we introduce three alternative methodologies for leveraging LLMs to achieve fair re-ranking in scenarios where the sensitive demographic attributes are missing. Each of the three proposed methodologies correspond to a similarly structured two-stage pipeline consisting of (i) data preparation followed by (ii) fair re-ranking.

### 4.1 Two-Stage LLM Fair Reranking Framework

We now describe the core component of two-stage pipeline referenced above shared by the three alternate strategies. The first stage prepares the inputs required for fairness-aware re-ranking, while the second stage of the pipeline operationalizes the fair re-ranking problem defined in Section 3. This overall structure is illustrated in Fig. 1.

#### 4.1.1 Data Preparation Stage

Since LLMs are designed to process natural language inputs (Vaswani et al. (2017)), the main goal of the first stage is to transform the structured tabular input data into a textual format consumable by LLM models. Building on recent methods from the literature, we employ the "List Template" serialization method introduced by (Hegselmann et al. (2023)) for this task. In this approach, each data row in the input list is systematically converted into a structured list of column names paired with their corresponding values. Lists candidates are indexed to enhance readability and help the LLM delineate individual records.

For the strategy where explicit inferencing is explored, this stage may additionally include demographic attribute inference. Because group fairness relies on knowledge of the group membership of the candidates, one approach employs third-party inference services to estimate missing sensitive attributes based on available features. For example, sex may be inferred from a candidate's name and nationality.

#### 4.1.2 Fair Reranking Stage

By the end of the first stage, the sensitive demographic attributes either remain missing or inferred (some potentially incorrectly), and the data is serialized. In the second stage, we use an LLM as the re-ranking mechanism. Specifically, we use a unified prompt-based instruction template to guide the model toward fairness-aware reordering. The final output is a re-ranked list that aims to balance utility and fairness.

Our *prompt design*, used in all three approaches, is a multi-layered template-based approach to guide toward generating fair and well-structured ranked outputs. At its core, the prompt consists of a fairness instruction

---

[3]https://gender-api.com/
[4]https://namsor.app/

along with the serialized input list to turn it into a fair ranking. In few-shot settings, example input-output pairs of utility-ranked lists (input) and their corresponding fairly re-ranked ordering (output) are incorporated to demonstrate the desired behavior. The prompt template includes five core components:

1. *Pre-Prompt:* This is the task context, defining the LLM's role and goals (e.g., ranking officer, loan officer). It is included as a system prompt for instruction-tuned models like `Llama-Instruct`.

2. *Fairness Instruction:* The fairness instruction is a statement that provides details on the attribute to be considered by the LLM when incorporating fairness (e.g., sex, race):

   > `Re-rank the following list to incorporate fairness with respect to {attribute}.`

   We deliberately do not instantiate a specific fairness definition, allowing us to evaluate how LLMs respond to a notion of fairness as a basic signal.

3. *Serialized Input List:* Depending on the strategy, the input may include names, scores, and optionally the inferred sensitive attributes.

4. *Few-Shot Examples:* Example rankings are formatted as lists of $n$ candidates, where the input is sorted by utility and the output reflects fair re-ranking generated using a standard fair ranking algorithm such as DetConstSort (Geyik et al. (2019)). Only examples yielding high fairness scores (based on exposure ratio) are utilized, ensuring quality guidance.

5. *Post-Prompt:* This final component guides the LLM in structuring its output in the expected format.

Building on this representation, we construct the input prompt $p$ based on its components as:

$$p = P \left[ \oplus \mathcal{S} \right] \oplus T \oplus \texttt{TabSerialize}(C, a) \oplus O, \tag{2}$$

where $\oplus$ denotes concatenation, $P$ is the pre-prompt, $\mathcal{S}$ represents optional exemplar shots, $T$ is the fairness instruction, $\texttt{TabSerialize}()$ denotes the table-to-text serialization of the data $C$ with attributes $a$,[5] and $O$ is the post-prompt.

### 4.2 Approach 1: LLM Fair ReRanking Using Attribute Inference via Third-Party Service

This approach, seen in Fig. 2a, conditions the LLM's re-ranking stage on externally inferred sensitive attributes obtained during data preparation. A structured prompt example is shown in Fig. 17 (Appendix Section A.13).

**Data Preparation Stage.** Missing sensitive attribute values are estimated using *traditional demographic inference services,* such as, Behind-the-Name, Gender-API, etc (see Section 3.3),

We evaluate three widely used name-based sex inference services, which, as we show in Section 5, exhibit differing levels of errors. We represent the inferred demographic attribute for a candidate $i$ as $\tilde{a}_i = g(x_i)$, and the list of inferred attributes for all candidates in $C$ is denoted $\tilde{a}$. These inferred attributes are incorporated into the serialized representation defined in Section 4.1. The resulting prompt is therefore expressed in terms of $\tilde{a}$ rather than ground-truth value attributes:

$$p = P \left[ \oplus \mathcal{S} \right] \oplus T \oplus \texttt{TabSerialize}(C, \tilde{a}) \oplus O. \tag{3}$$

**Fair Reranking Stage** The LLM receives the serialized list augmented with inferred attributes and utility scores. The prompt explicitly discloses that demographic attributes were inferred rather than ground-truth (see Fig. 17). Optionally, we include in the prompt few-shot examples (shots) consisting of names, ground-truth attributes, and corresponding scores. The examples are presented as input-output pairs, illustrating representative instances of typical inputs alongside their corresponding outputs.

---

[5]Note that the demographic attribute vector is $a$, the individual attribute values are $a_i$ and the inferred demographic attribute vector is represented as $\tilde{a}$

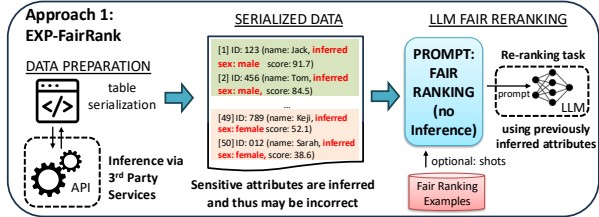

(a) Approach 1 (EXP): LLM Fair Re-Ranking Using Attribute
Inference via Third-Party Service

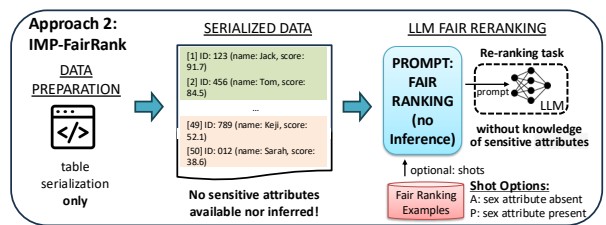

(b) Approach 2 (IMP*): LLM-Based Fair Re-Ranking without
Sensitive Attribute Inferencing

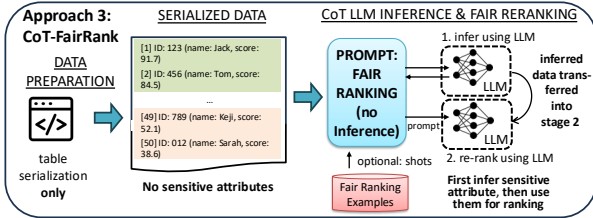

(c) Approach 3 (CoT): CoT-FairRank

Figure 2: Proposed approaches for LLM-based fair re-ranking with missing sensitive attributes

## 4.3 Approach 2: LLM-Based Fair ReRanking without Sensitive Attribute Inferencing

This approach, seen in Fig. 2b does not rely on support from external cues or explicit demographic inference
instruction. Instead, it requires the LLM to perform such inference implicitly in order to incorporate fairness.
A structured prompt example is illustrated in Fig. 18 (Appendix).

**Data Preparation Stage.** We convert the input table to a serialized list without performing demographic
inference (Fig. 2b). The attributes in their serialized format are thus: ID, name, and the score. Hence $\tilde{a}$ is
absent in $p$ (see Eq. 2).

**Fair Reranking Stage** The LLM receives only IDs, names, and utility scores, requiring it to rely on
internal signals to perform fairness-aware re-ranking (Fig. 2b). Prompts do not reference demographic
attributes or inference. We consider *two few-shot variants*. IMPLICIT-A, where the sensitive attributes
are absent from the examples and IMPLICIT-P, where ground-truth attributes are present in the examples.
These variants assess performance under fully unassisted conditions versus settings where examples implicitly
convey the role of demographic information in fair re-ranking.

## 4.4 Approach 3: Chain-of-Thought Inferencing and Fair ReRanking via LLM

This approach, depicted in Fig. 2c, assesses whether instructing the LLM explicitly to first infer the missing
attributes and then to use those results for fair re-ranking improves fairness outcomes. To model this, we
utilize Least-to-Most (Zhou et al. (2023)), an enhanced form of the Chain-of-Thought style of prompting.
An example prompt is illustrated in Fig. 19.

**Data Preparation Stage** Same as in Approach 2.

**Fair Reranking Stage.** In the fair re-ranking stage, the LLM is explicitly guided to perform a structured
multi-step reasoning process in a chain-of-thought format (illustrated in Fig. 19 from Appendix A.9). First, it
is instructed to infer the sensitive attribute from the candidates' names. Thereafter, it is instructed to utilize
these inferred attributes to conduct fairness-aware re-ranking. The initial input (in the first step) includes
only names and utility scores. In the second step, the LLM-inferred attributes are explicitly reintroduced
into the input to perform the final re-ranking step.

| | Baselines | Approach 1 (EXP) | | Approaches 2 (IMP*) & 3 (CoT) | |
|---|---|---|---|---|---|
| Initial | ranked by ground-truth scores | GT | using ground-truth sens. attribute | IMP | (App #2) zero-shot |
| | | GAPI | with Gender API | IMP-P | (App #2) sens. attr. present in shots |
| DCS | ranked by DetConstSort | BTN | with Behind the Name | IMP-A | (App #2) sens. attr. absent in shots |
| | | NMS | with Namsor | CoT | (App #3) chain-of-thought |
| | | ETHCNN | with EthCNN | | |
| | | Ethnicolr | with Ethnicolr | | |

Table 1: Fair ranking strategies, including baselines. Legend: EXP: Explicit-FairRank, IMP*: Implicit-FairRank, CoT: CoT-FairRank.

# 5 Design of our Experimental Study

**Datasets.** We select ranking datasets in which measured fairness is low under standard ranking procedures, making them appropriate for evaluating fairness-aware re-ranking under demographic uncertainty. We utilize four datasets, each containing sensitive attributes such as sex or race:

- STARTUP FOUNDERS: This dataset is derived from publicly available records of U.S. startup founders who raised Series A funding, compiled from Crunchbase. We use the aggregated Series A funding amounts as reference (groundtruth) scores for ranking founders. In this dataset, race is used as the sensitive attribute and is modeled as a multivalued attribute with four categories.[6] Dataset group statistics: Asian (Far East, Southeast Asia, and the Indian subcontinent): 584, Black/African: 42, Hispanic/Latino: 63, White/Caucasian (Non hispanic): 2637.

- (W)NBA: This dataset captures performance statistics from elite basketball players across the NBA and WNBA. We use the all-time career points up to 2017 as the reference scores. Dataset group statistics: Male: 3763, Female: 953.

- BOSTONMARATHON: Collected from the 2017 Boston Marathon, this dataset provides stage-wise performance data for runners across sex categories. We use the 'Official Time' column for our reference scores. Times are normalized such that the longest time is given the lowest score to have an appropriate relevance-ordered ranking set [7]. Dataset group statistics: Dataset group statistics: Male: 14438, Female: 11972.

- COMPAS: This dataset originates from a public investigation into recidivism risk assessments used in the U.S. justice system involving the COMPAS algorithm (Larson et al. (2016)). We use the recidivism (*likelihood to re-offend*) scores as reference scores. Dataset group statistics: Male : 9336, Female : 2421.

**Test set Preparation.** Each data set is split into 80% for *curating the few-shot examples* (detailed below) and 20% for *testing*. To identify the disadvantaged group, we plot skew graphs (Section 5.1) and observe which group exhibits lower representation at the top. For test set generation, we sample, from the test split, 10 subsets of candidates each to account for variance, ensuring that the relative skews in group representation mirrors that of the original test set (Section 5.1). To best understand the effect of re-ranking, each group defined by the sensitive attribute is equally represented in the subsets. Specifically, in the gender-based experiments, there are 25 female and 25 male instances, while in the race-based experiments, each racial group contains 25 instances.

---

[6]We recognize that representing race as a fixed set of categories is inherently limited and may obscure social, cultural, and individual complexity.

[7]https://www.kaggle.com/datasets/rojour/boston-results?resource=&select=marathon_results_2017.csv

| Model (shorthand) | Official model name | Access method |
|---|---|---|
| Gemini | `gemini-2.0-flash-thinking-exp-01-21` | Gemini API (free) |
| | `gemini-3-flash-preview` | Gemini API (paid) |
| DeepSeek | `deepseek-chat` (DeepSeek-V3) | Paid API (also available for free via |
| | `DeepSeek-V3.2` (used for race-based experiments) | Hugging Face) |
| LLaMA | `Llama-3-8B-Instruct` | Hugging Face transformers (open-source) |

Table 2: Large language models used in our experiments, including shorthand references, official model names, and access modalities.

**Creating Example Shots.** Using the 80% portion of the datasets, we sample four distinct subsets and apply the fairness-aware re-ranking algorithm, DetConstSort (Geyik et al. (2019)) to each. This process yields fairly re-ranked sets, which we use as few-shot examples in our experiments. To ensure that the shots are expressive, we constrain the average exposure ratio to be close to 1 (Section 5.1).

**Comparative Fair Ranking Baseline Methods.** We evaluate our approach against two traditional baselines. The first baseline corresponds to the original utility-based rankings, allowing us to assess the trade-off between utility and fairness introduced by our method. The second baseline employs DetConstSort (Geyik et al. (2019)), a widely-used fairness-oriented re-ranking algorithm based on traditional machine learning techniques, to generate fairer rankings. Additionally, for our first LLM-based strategy (Approach 1), we include an extra set of baselines where the re-ranking is guided by ground-truth demographic attributes to benchmark the performance of inference-based fairness adjustments.

**LLMs used as Backbone Models.** We experiment with three LLMs selected for their diversity and relevance to the task (Table 2), namely, open-source models—DeepSeek and LLaMA—and proprietary model, Gemini. DeepSeek and LLaMA were accessed using paid production-grade services; while we experimented with both the freely available version and the paid API access of Gemini. We use few-shot prompting as a reproducible, model-agnostic means of controlling model behavior across backbones and versions.

**Third-Party Attribute Inference and Treatment of Unknowns.** We experiment with three popular services for inferring missing demographic attributes (see Section 3.3). In our study, we experiment with the sensitive attribute being inferred given a person's name as inherent component of Approach 1 (Section 4.2). Each inference service returns candidates whose sensitive attribute could not be determined with a degree of certainty above a threshold, hereby referred to as 'unknowns'. We addressed this by following the steps in Olulana et al. (2024) for the gender experiments. Table 1 collects all strategies and their abbreviated names used in this paper, including baselines. For the race inference, we exclude candidates labeled as unknown from the analysis. We adopt this strategy to avoid introducing an additional heterogeneous group whose composition and size depend on inference confidence and may itself correlate with race. For this, we follow the approach previously used in the literature, including by Ghosh et al. (2021). Treating unknowns as a separate group in the race setting could confound fairness evaluation and obscure interpretation of race-based disparities. We therefore restrict the race experiments to candidates with confidently inferred race labels. We note the exclusion of candidates with unknown race as a limitation of this part of the study. In particular, it is possible that the removed cases may correspond to less common or underrepresented names, which can itself reflect disparities in the inference system. Future work should investigate fairness evaluation methods that explicitly account for uncertainty or partial demographic labels rather than relying on exclusion.

## 5.1 Evaluation Metrics

To assess fair rankings, we use three fairness metrics—two based on representational parity (Skew and NDKL) and one on exposure ratio—and a standard utility metric.

### 5.1.1 Skew (Geyik et al. (2019)).

Skew quantifies the relative visibility of a demographic group within the top prefix of a ranking. For a given group $g \in G$, the skew at position $k$ in a ranking $\tau$ is computed as:

$$Skew@k(\tau)_g = \frac{p_{\tau@k,g}}{p_{C,g}}, \tag{4}$$

where $p_{\tau@k,g}$ is the share of group $g$ in the top $k$ ranked candidates, and $p_{C,g}$ is the group's proportion across the full candidate set $C$.

### 5.1.2 NDKL (Geyik et al. (2019)).

The Normalized Discounted Kullback-Leibler divergence measures the divergence between the group distribution in the top $k$ results and that of the entire candidate set $C$, with a logarithmic discount applied to account for position relevance:

$$NDKL@k(\tau) = \frac{1}{Z} \sum_{i=1}^{k} \frac{1}{\log_2(i+1)} d_{KL}(P_{\tau@k}||P_C), \tag{5}$$

where $d_{KL}(P_{\tau@k}||P_C)$ is the KL-divergence between group distributions in the top-$k$ candidates and the full set $C$ and $Z = \sum_{i=1}^{k} \frac{1}{\log_2(i+1)}$ is a normalization constant.

### 5.1.3 Normalized Discounted Cumulative Gain (NDCG) (Järvelin and Kekäläinen (2002)).

NDCG, a standard metric for quantifying ranking quality based on candidate relevance, is defined as:

$$NDCG@k(\tau) = \frac{1}{Z} \sum_{i=1}^{k} \frac{s_i}{\log_2(i+1)}, \tag{6}$$

where $s_i$ is the relevance score assigned to the candidate at position $i$ and $Z$ is a normalization factor ensuring the metric ranges from 0 to 1. A perfect ranking (where the highest-relevance candidates appear first) yields NDCG = 1.

## 6 Experimental Study: Results and Analysis

In this section, we analyze LLM-based fairness-aware re-ranking across four settings: demographic inference from names (§6.1), fair ranking with ground-truth attributes (§6.2), re-ranking under missing demographic information (§6.3), and the impact of few-shot examples on integrated inference and re-ranking (§6.4), and the effect of backbone LLM choice on fairness, utility, and stability without external inference (§6.5). Throughout this section, we distinguish between two fairness settings that differ in difficulty. The binary-sex setting (Boston Marathon, (W)NBA, COMPAS) involves a two-group attribute where inference is relatively tractable and fairness interventions target a single protected/non-protected split. The multi-group race setting (Startup Founders) involves four racial groups, finer-grained inference, and a correspondingly more challenging fairness objective. We therefore report results separately for these settings throughout, as they exhibit systematically different behavior, a pattern that becomes especially visible in the statistical analysis in Section 6.6.

### 6.1 Inference Accuracy

Given that fair re-ranking inherently relies on access to the group membership of each candidate, we first study the *accuracy of the sensitive attribute inference* that serves as a core component of our fair re-ranking methods. For this, we evaluate how accurately LLMs infer sensitive attributes, and compare this with the performance of the third-party attribute inference services (**RQ1 & RQ2**)[8]. LLM-based attribute inference is a core component of our CoT-FairRank strategy (Approach 3), while third-party inference services are utilized by EXP[9] (Approach 1). IMP performs ranking without attribute inference, and is thus discussed later in Section 6.3. While name-based demographic inference risks reproducing cultural and linguistic biases, these methods have prevalence in real-world systems. Our goal is not to endorse these practices, but instead to evaluate how commonly deployed inference tools interact with fairness-aware re-ranking in practice and to surface their limitations and risks. *Note that for each dataset, we conduct a memorization test to ensure that inference results by the LLMs are not due to memorization (see Appendix A.11).*

---

[8]Additional results on RQ1 will be presented in Section 6.3.

[9]The names of the inference services are also used to refer to the LLM-based methods that employed them as third-party inference components.

| | Inference Method | Dataset and Sensitive Attribute | | | |
|---|---|---|---|---|---|
| | | Startup Founders (race) | Boston Marathon (sex) | (W)NBA (sex) | COMPAS (sex) |
| *Third-party services (Approach 1)* | Gender API | – | 98% | 96% | 99% |
| | Behind The Name | – | 76% | 75% | 84% |
| | Namsor | – | 40% | 41% | 40% |
| | EthCNN | 40% | – | – | – |
| | Ethnicolr | 37% | – | – | – |
| *Approach 2* | Not Applicable: Does not perform explicit attribute inference by definition. | | | | |
| *LLMs (Approach 3, 0-shot)* | Gemini | 73% | 98% | 100% | 94% |
| | DeepSeek | 55% | 56% | 77% | 88% |
| | Llama | 57% | 63% | 79% | 93% |

Table 3: Inference accuracy across datasets used in the experiments. Race inference is evaluated on the Startup Founders dataset, while gender inference is evaluated on the Boston Marathon, (W)NBA, and COMPAS datasets. Third-party services correspond to Approach 1 (EXP), while LLM-based inference corresponds to Approach 3 (CoT-FairRank); sensitive attribute corresponding to each dataset.

Table 3 reports the inference accuracy yielded by third party inference services and by the inference step in CoT-FairRank for each LLM backbone model, on the test sets of the three benchmark datasets. The wide variation in demographic inference accuracies provides a robust foundation for evaluating performance across different tools with varying capabilities in demographic inference.

**Sex inference (binary).** Gemini reaches 94–100% accuracy on all three sex datasets, rivaling GenderAPI. DeepSeek and Llama cluster in the 56–93% range. Note specifically that (W)NBA is the easiest (both LLMs >77%) while Boston Marathon is hardest for non-Gemini models, a point worth speculating on (possibly because marathon data includes more international names with ambiguous gender signal).

**Race inference (4-group).** Race inference is substantially harder than sex inference across all methods. The best-performing LLM (Gemini, 73%) substantially outperforms the best third-party service (EthCNN, 40%), but remains well below the near-ceiling accuracies achieved on sex. This asymmetry shapes every downstream result: fairness interventions that work cleanly for sex frequently degrade for race, as we show in Section 6.3.

## 6.2 Fair Re-Ranking with Ground Truth

Next, we investigate the viability of LLMs as re-ranking models. To isolate this capability from their demographic inference performance, we conduct the evaluation using ground-truth demographic attributes. While this does not address any specific research question, it provides an important foundation for employing LLMs in the fair re-ranking task. Specifically, we use race for the Startup Founders dataset and sex for the Boston Marathon, (W)NBA, and COMPAS datasets.

Fig. 3 shows NDKL and NDCG obtained by the different LLMs for the Startup Founders and Boston Marathon datasets when ranking is performed with access to the ground-truth demographic attributes. Across both datasets, we observe a consistent reduction in NDKL values relative to the Initial ranking, indicating improved fairness, except for Llama on the Startup Founders dataset, where the fairness metric slightly worsens. Gemini consistently outperforms the baseline fairness re-ranking algorithm (DCS) in terms of NDKL. DeepSeek performs competitively with DCS and in some cases surpasses it, particularly on the Boston Marathon dataset. Llama also improves fairness relative to the Initial ranking, although its performance remains inferior to the DCS baseline for the Boston Marathon dataset.

**On sex (Boston Marathon).** NDKL drops sharply (from 0.22 to 0.06 for Gemini, and to 0.08 for DeepSeek), and all three LLMs match or outperform DCS. This suggests LLMs can execute binary fairness interventions reliably when ground truth sensitive attribute is provided.

**On race (Startup Founders).** Improvements are smaller and less consistent. Gemini reduces NDKL from 0.21 to 0.10, but Llama *worsens* fairness (NDKL rises to 0.22). The multi-group setting exposes model-dependent behavior that the binary setting masks. This foreshadows another finding to be discussed in Section 6.5, namely that Llama is systematically weaker—a weakness that has not been observed in the binary attribute case.

These results demonstrate the LLMs' promising capacity for fair re-ranking. However, the observed inconsistencies motivate exploration of additional approaches to determine whether the performance of these LLM-based solutions can match or exceed the baseline traditional algorithm. In the following sections, we focus on the results obtained from DeepSeek and Gemini, as these models showed stronger and more consistent performance than Llama in our experiments.

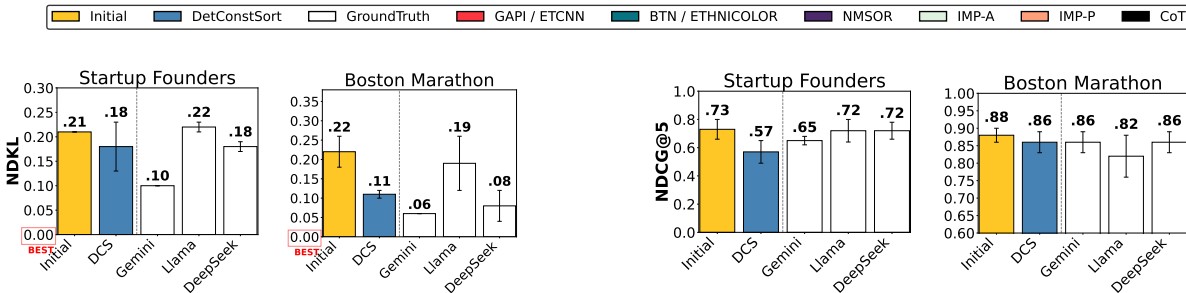

Figure 3: Fairness metrics—NDKL (left) and NDCG (right)—for LLM-based fair reranking methods evaluated across datasets. Legend is the same for all Figures.

### 6.3 Zero-shot Strategies with Uncertain Attributes

We evaluate fairness-aware re-ranking under demographic uncertainty by comparing a traditional baseline, DetConstSort (DCS), with zero-shot variants of the LLM-based approaches introduced in Section 4. We compute NDKLs and NDCGs on all four datasets. Fig. 4 shows NDKL (top) and NDCG (bottom) obtained on the STARTUP FOUNDERS dataset; using Gemini (two left-most columns) and DeepSeek (two right-most columns) as backbones. Fig. 4 presents results side-by-side on a race dataset (Startup Founders) and a sex dataset (Boston Marathon), using Gemini (two left-most columns) and DeepSeek (two right-most columns) as backbones; this pairing allows us to observe how attribute difficulty relates to the three approaches. Results for the remaining sex datasets ((W)NBA, COMPAS) are reported in the Appendix (Sections A.6 and A.7).

The Approach 1 (EXP third-party services-based) results are ordered based on the quality of the third-party attribute inference services used, namely, from most (GAPI) to least accurate (NMSOR). Across all settings, the LLM-based approaches (GT, GAPI, IMP, and CoT-FairRank) achieve lower representation disparity (NDKL) than DCS, except when external demographic inference is highly inaccurate (BTN, NMSOR). All variants improve over the Initial ranking, even without exemplar guidance, suggesting that LLMs can exhibit fairness-aware behavior without task-specific tuning.

The three approaches behave similarly on sex: all tend to improve fairness over Initial, with CoT often performing best (Fig. 4, Boston Marathon panels). On race, the pattern is less uniform (Fig. 4, Startup Founders panels). External inference (Approach 1) is limited by the 37–40% accuracy of third-party race classifiers, implicit inference (Approach 2) shows smaller improvements, and CoT tends to perform best with Gemini, a pattern that aligns with our finding in Section 6.1 that Gemini's self-inference (73%) is more accurate than any external race classifier we evaluated. Taken together, these results suggest that LLM-based fairness methods may be most useful in settings where external tools perform worst. In addition, we observe that:

- *Approach 1 (EXP).* When inferred sensitive attributes are accurate or have low error rates (e.g., via GAPI; see Section 5), fairness improves relative to the initial ranking, as reflected by lower NDKL values. In contrast, higher inference error rates (e.g., with NMSOR) degrade fairness, leading to worse NDKL

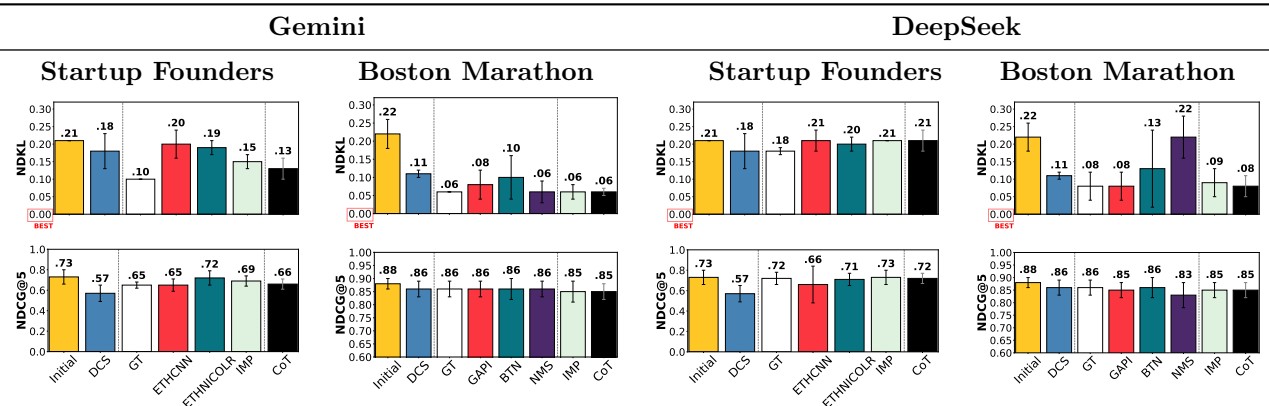

Figure 4: Fairness (NDKL) and Utility (NDCG) for LLM-based reranking on the STARTUP FOUNDERS and BOSTON MARATHON datasets.

outcomes; the same pattern is observed for AveExpR (see Section A.2 in Appendix). This degradation arises because misclassified attributes misguide the re-ranking process, causing the fairness objective to be optimized with respect to incorrect groups. The effect is amplified for race, where all three external services fall below 40% accuracy (Table 3); Approach 1 on Startup Founders is therefore constrained by the accuracy ceiling of available classifiers. Accordingly, in addressing **RQ1**, external demographic inference services appear most effective when their predictions are sufficiently accurate; otherwise, they may risk undermining fairness rather than improving it.

- *Approach 2 (IMP).* We observe consistent improvements in NDKL relative to the Initial ranking, with performance exceeding the traditional fair re-ranking algorithm (DCS) and, under conditions of perfect (100%) inference accuracy, matching that of Approach 1. For AveExpR (Section A.2 in Appendix), IMP likewise improves upon the Initial ranking and reaches parity with Approach 1 when inference is error-free. These gains are more pronounced on sex than on race: IMP approaches the ground-truth condition on Boston Marathon, while a larger gap to ground-truth remains on Startup Founders (Fig. 4). In addressing **RQ3**, these results indicate that LLMs can, in some settings, serve as the inference mechanism within fair re-ranking pipelines.

- *Approach 3 (CoT).* This approach yields further improvements in NDKL over both the Initial ranking and DCS. For AveExpR (Section A.2 in Appendix), we also observe consistent gains relative to Initial, reinforcing the advantage of designs that leverage LLMs for inference and thereby providing additional support for **RQ3**. CoT's advantage over IMP is clearest on race, where Gemini's self-inference (73%) exceeds every external race classifier; on sex, where both IMP and external inference already perform well, the additional gains from CoT are smaller. Notably, CoT consistently outperforms IMP, suggesting that within end-to-end LLM frameworks, chain-of-thought reasoning is the preferred may be a more effective strategy.

In summary, LLM-based approaches appear promising: fairness improves upon Initial's, while utility remains relatively stable. A robustness analysis varying the proportion of the disadvantaged group (Appendix A.12, Fig. 22) confirms that these fairness gains hold across imbalanced settings, with the gap between fair and Initial rankings narrowing only at extreme proportions (80–90%).

## 6.4 Effect of Shots on Fair LLM-Based Re-Ranking Strategies With Uncertain Demographic Attributes

Next, we investigate how the inclusion of few-shot exemplars of fairly ranked lists affects the outcomes of LLM-based re-ranking, i.e., whether they improve the model's ability to balance fairness with utility (**RQ4**). As a side objective, we study the extent to which providing shots can mitigate the effects of varying error levels of the inferred demographic information.

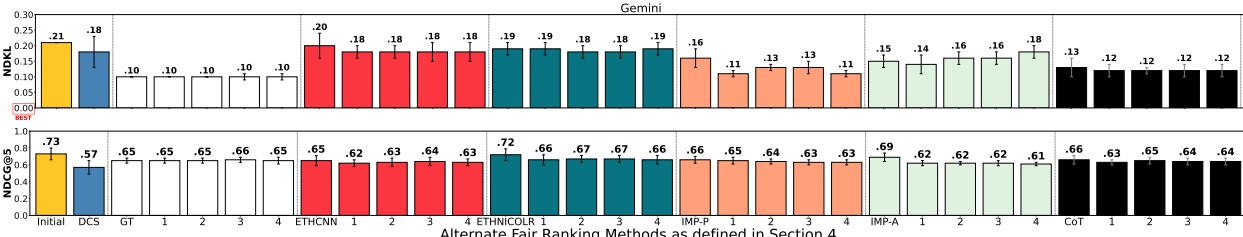

Figure 5: Fairness (NDKL and Average Exposure) and Utility (NDCG) metrics for fair reranking results varying the number of shots using Gemini on STARTUP FOUNDERS dataset.

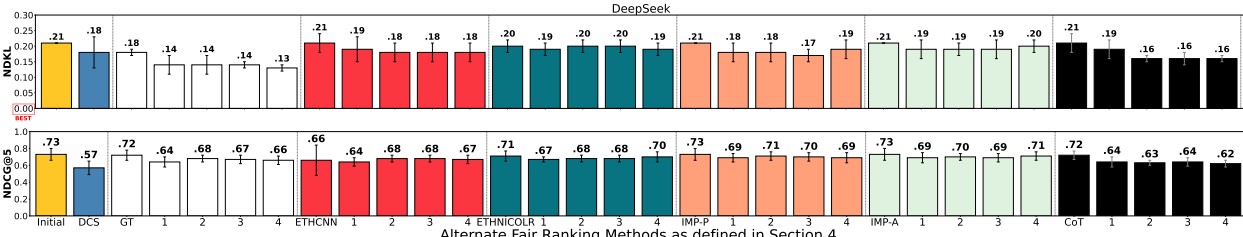

Figure 6: Fairness (NDKL and Average Exposure) and Utility (NDCG) metrics for fair reranking results varying the number of shots using DeepSeek on STARTUP FOUNDERS dataset.

We report results for Gemini and DeepSeek on the STARTUP FOUNDERS (Figs. 5, 6) and BOSTONMARATHON (Figs. 7, 8) dataset in this section; [10] results for additional datasets are provided in the Appendix (Figures 14, 15 and 16). For third-party inference services with high accuracy (Approach 1), adding exemplar shots consistently moves the results closer to those of the fair re-ranking baseline, DCS The red and green bars in Figure 7). When inference error is negligible (GT, GAPI), few-shot prompting yields NDKL values comparable to DCS (white and red bars in Figure 7). In contrast, when inference quality is poor (NMSOR), the fairness signal conveyed in the few-shot examples are less effectively reflected in the outputs, likely due to inaccuracies in demographic inference, and do not yield clear improvements in re-ranking effectiveness (the green and red bars in 5 and the purple bars in 7). Approaches IMP-A, IMP-P, and CoT-FairRank also benefit from exemplar shots, with CoT-FairRank achieving DCS-level fairness using fewer examples (orange, green and black bars on Figures 5, 6 and 8).

Crucially, the benefit of shots is larger on Startup Founders (race, 4 groups) than on Boston Marathon (sex, binary): on race, 2-shot configurations move NDKL closer to the DCS baseline than any 0-shot configuration achieves, whereas on sex the 0-shot results are already near the achievable floor. This attribute-dependent shot sensitivity foreshadows the statistical findings in Section 6.6. Overall, these results suggest that few-shot prompting can guide LLM-based re-ranking to match dedicated fairness algorithms while largely preserving utility, addressing **RQ4**.

---

[10]Similar results are reflected on the other datasets. See Appendix - (Figures 14, 15 and 16)

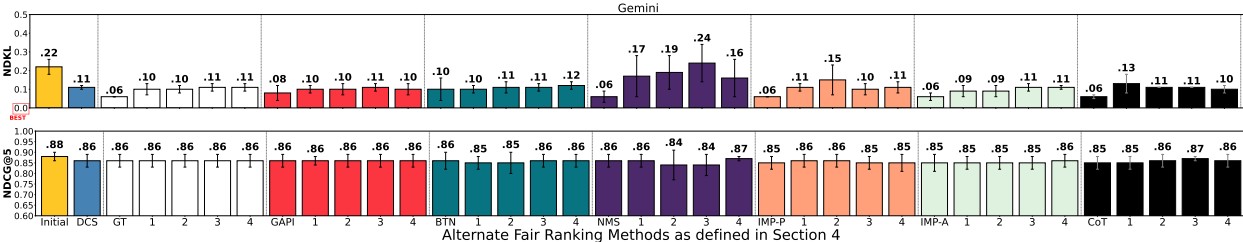

Figure 7: Fairness (NDKL and Average Exposure) and Utility (NDCG) metrics for fair reranking results varying the number of shots using Gemini on BOSTON MARATHON dataset.

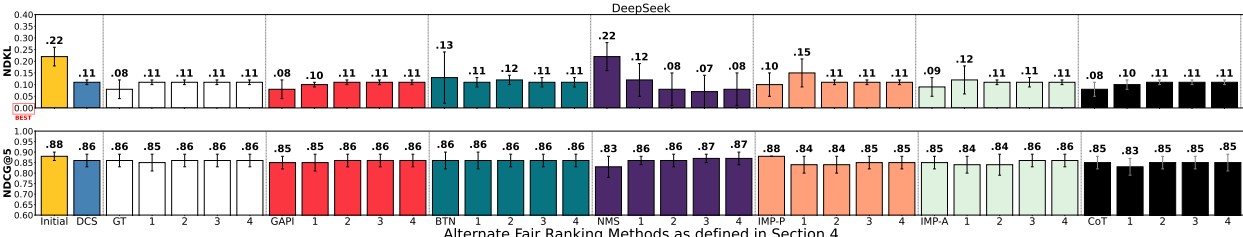

Figure 8: Fairness (NDKL and Average Exposure) and Utility (NDCG) metrics for fair reranking results varying the number of shots using DeepSeek on BOSTON MARATHON dataset.

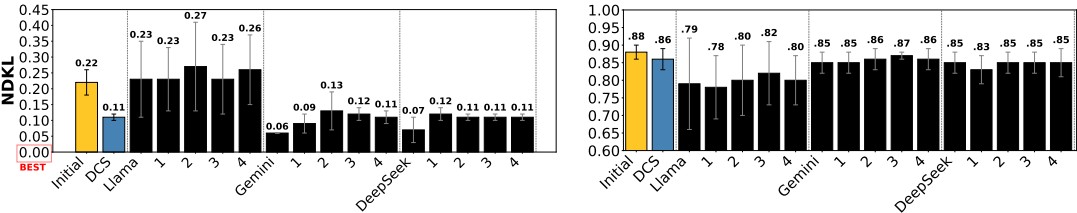

Figure 9: Fairness (NDKL) and Utility (NDCG) for CoT-FairRank across LLMs with shots varying from 0 to 5 on BOSTONMARATHON.

## 6.5 Comparison of Alternate Backbone LLMs for Fair Re-Ranking

In this section, we examine the impact of different LLM backbone models (Gemini, DeepSeek, and Llama) on the effectiveness and stability of LLM-based fair re-ranking, and in particular, the CoT-FairRank, IMP-A, and IMP-P approaches. These models operate without external inference services, representing scenarios where LLMs receive no assistance regarding demographic attributes. This analysis directly builds upon our investigation of **RQ4**, showing how backbone choice directly impacts fairness–utility outcomes under demographic uncertainty. Fig. 9 and, from Appendix A.10, Figures 20 and 21 respectively present the results of CoT-FairRank, IMP-A and IMP-P, evaluated across three LLMs. Across the three strategies, Gemini and DeepSeek perform comparably, with the inclusion of shots consistently reducing variability and improving effectiveness. Notably, DeepSeek reaches stability and near-optimal results after about the third shot, while Gemini follows closely with

similar performance.

## 6.6 Statistical Significance and Variance Analysis Across Models, Approaches and Shots.

To strengthen the statistical rigor of our analysis, we conducted a targeted evaluation of the fairness and utility metrics, NDKL and NDCG@5 (Figure 10). Specifically, we performed paired Wilcoxon signed-rank tests comparing 0-shot and few-shot configurations (0-shot, 1-shot, and 2-shot) against the Initial ranking, in order to assess whether zero-shot prompting is sufficient or whether the inclusion of few-shot exemplars yields statistically meaningful differences. We evaluate both the BOSTON MARATHON dataset, which uses binary sex as the sensitive attribute, and the Startup Founders dataset, which uses race across four groups, a more demanding setting given the finer-grained attribute space and the generally lower inference accuracy of the inference-based methods (EthCNN and Ethnicolor) on race compared to sex (with the exception of NMSOR, which performed poorly on sex inference).

BOSTON MARATHON. NDCG@5 differs significantly from the Initial ranking across all methods, shot configurations, and LLM families, indicating that every re-ranking approach produces utility shifts that are distinguishable from the baseline. The NDKL results reveal a more nuanced, model-dependent pattern. At 0-shot, DeepSeek and Gemini exhibit significant differences from Initial across all methods except NMSOR, consistent with NMSOR's lower inference accuracy on sensitive attributes. Llama shows the opposite pattern: only NMSOR differs significantly from Initial, while all other methods are statistically indistin-

guishable from the baseline, suggesting that 0-shot prompting is insufficient to elicit fairness-aware behavior from Llama. At 1-shot, most configurations yield statistically significant differences with respect to Initial, both for NDCG@5 and NDKL. Notably: NMSOR remains non-significant; DeepSeek and Llama have some additional configurations where the NDKL difference is not statistically significant. At 2-shots, we again note that NDKL of rankings that used NMSOR for inference are consistently indistinguishable from Initial. Fewer configurations led to non-statistically significant differences in fairness than at 1-shot. CoT, however, fails to produce a significant NDKL difference from Initial for Llama at any shot configuration, indicating that chain-of-thought prompting alone does not shift Llama's ranking behavior on this dataset.

Startup Founders. The Startup Founders results show a similar but more pronounced dependence on shot count, likely reflecting the added difficulty of reasoning over four racial groups rather than binary sex. At 0-shot, significance is fragmented across methods and LLM families: Llama in particular shows non-significant NDKL differences for ethcnn, ethnicolor, and IMP, with only CoT producing a significant fairness shift. DeepSeek at 0-shot shows a striking p-value of 1.000 for IMP on NDKL, indicating no detectable difference from the Initial ranking, and CoT fails to shift either metric significantly. Gemini is the most responsive family at 0-shot, though it still shows non-significance for Ethnicolor on NDCG@5. At 1-shot, results sharpen considerably: nearly all method-family combinations reach significance on both metrics, with isolated exceptions (IMP-A on DeepSeek, EthCNN on Llama's NDKL). By 2-shots, the pattern consolidates further, every method across every LLM family produces a statistically significant difference from the Initial ranking, with the single exception of IMP-P on Llama's NDKL. This near-uniform significance at 2-shots is notable given that the inference-based methods (EthCNN and Ethnicolor) are known to have lower accuracy on race than on sex, and suggests that in-context examples can partially compensate for weaker sensitive-attribute inference when sufficient demonstrations are provided.

Taken together, these results suggest that shot configuration plays a meaningful role in fairness-aware ranking, with its importance amplified in more demanding settings. For BostonMarathon, Llama benefits most visibly from additional shots, while DeepSeek and Gemini already respond to fairness-aware prompting at 0-shot. For Startup Founders, where the sensitive attribute is race with four groups and inference accuracy is lower, the role of shots becomes more central across all three LLM families, with 2-shot configurations producing near-uniform significance. This is consistent with our analysis in Section 6.4, where increasing the number of shots is associated with improved fairness outcomes. This suggests that shot count may warrant consideration as part of the intervention design rather than as a secondary tuning choice, and that its value grows with the complexity of the sensitive-attribute space.

Next, in Figure 11, we compare approaches against each other to assess whether they produce statistically distinguishable ranking outcomes. For NDCG@5, most pairwise comparisons on DeepSeek and Gemini are non-significant, indicating that utility is largely similar regardless of the method chosen. Llama departs from this pattern: CoT differs significantly from every other method on utility, while other pairs remain indistinguishable. For NDKL, the panels are visibly greener, with substantially more pairwise significance across all three LLM families, indicating that method choice matters more for fairness than for utility. NMSOR emerges as the most distinguishable method on DeepSeek and Gemini, while CoT again plays that role on Llama. Together, these results suggest that when utility is approximately preserved across methods, method selection is most consequential for fairness.

## 7 Discussion

Here, we reflect on our findings, connect them to deployment considerations, clarify our contributions, and discuss key implications, opportunities, and limitations.

**Connection to deployment** While we see promising results for LLM-based re-ranking when sensitive attributes are missing, translating these findings into deployable systems requires careful attention to when, how and under what assumptions such pipelines are appropriate. The choice between what approach to use should be governed primarily by the quality of available sensitive attribute information rather than by a priori preference for any single strategy. When high-accuracy inference is available for the demographic attribute of interest (e.g., Gender API for sex inference in our experiments), EXP-style (Approach 1) pipelines

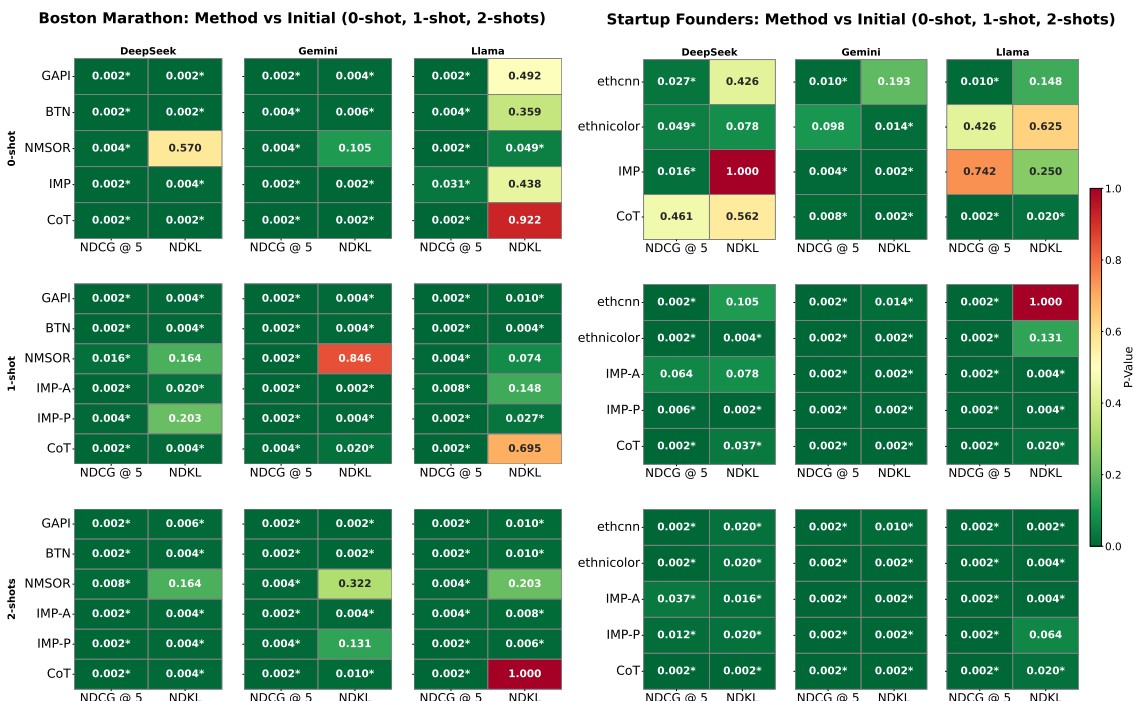

Figure 10: Paired Wilcoxon signed-rank tests comparing each method against the Initial ranking across shot configurations (0-shot, 1-shot, 2-shots) and LLM families (DeepSeek, Gemini, Llama). Cells report p-values, with asterisks (*) denoting statistical significance at at $\alpha = 0.05$.

remain a reasonable choice and align closely with DetConstSort-level fairness once few-shot exemplars are introduced. When inference quality is unknown, degraded, or unavailable, however, EXP pipelines risk actively harming fairness by optimizing re-ranking against misclassified groups. In such cases, CoT-FairRank (Approach 3) is the more defensible design: it consistently outperformed IMP (Approach 2) across datasets and backbones, and its explicit inference step offers a locus for auditability that implicit approaches lack. IMP remains attractive when exposing sensitive attributes — even as intermediate reasoning artifacts — is itself undesirable for privacy, legal, or organizational reasons, but practitioners and researchers should expect a modest fairness cost relative to CoT-FairRank.

In addition to approach selection, our results also highlight important practical considerations around backbone choice, prompting strategy, and exemplar design for fairness-aware LLM re-ranking. Backbone choice materially affects both performance and stability. In our experiments, Gemini and DeepSeek achieved stronger fairness–utility trade-offs than Llama, indicating that fair re-ranking is not model-agnostic and requires dataset-specific backbone evaluation. Few-shot prompting also improved consistency, suggesting that a small set of curated fair exemplars can enhance reproducibility. Future work should examine how pretraining and instruction tuning shape fairness-aware ranking behavior. Few-shot prompting consistently reduced the gap between LLM methods and dedicated fairness algorithms, with CoT-FairRank reaching near-DCS fairness using few exemplars. For deployment, exemplar sets should be carefully constructed and documented, as they become a key component of system behavior. However, few-shot guidance does not correct poor sensitive attribute inference, suggesting exemplars function as refinement rather than correction.

**Empirical Findings Under Demographic Uncertainty.** This work examined whether LLMs can support fairness-aware re-ranking when sensitive demographic attributes are missing, uncertain, or unavailable. Across a range of experimental settings, our findings suggest that, in many cases, LLM-based re-ranking strategies can improve fairness outcomes, as measured by exposure- and representation-based metrics, while largely preserving ranking utility. That said, the effectiveness of different strategies appears to vary depending on how demographic signals are incorporated.

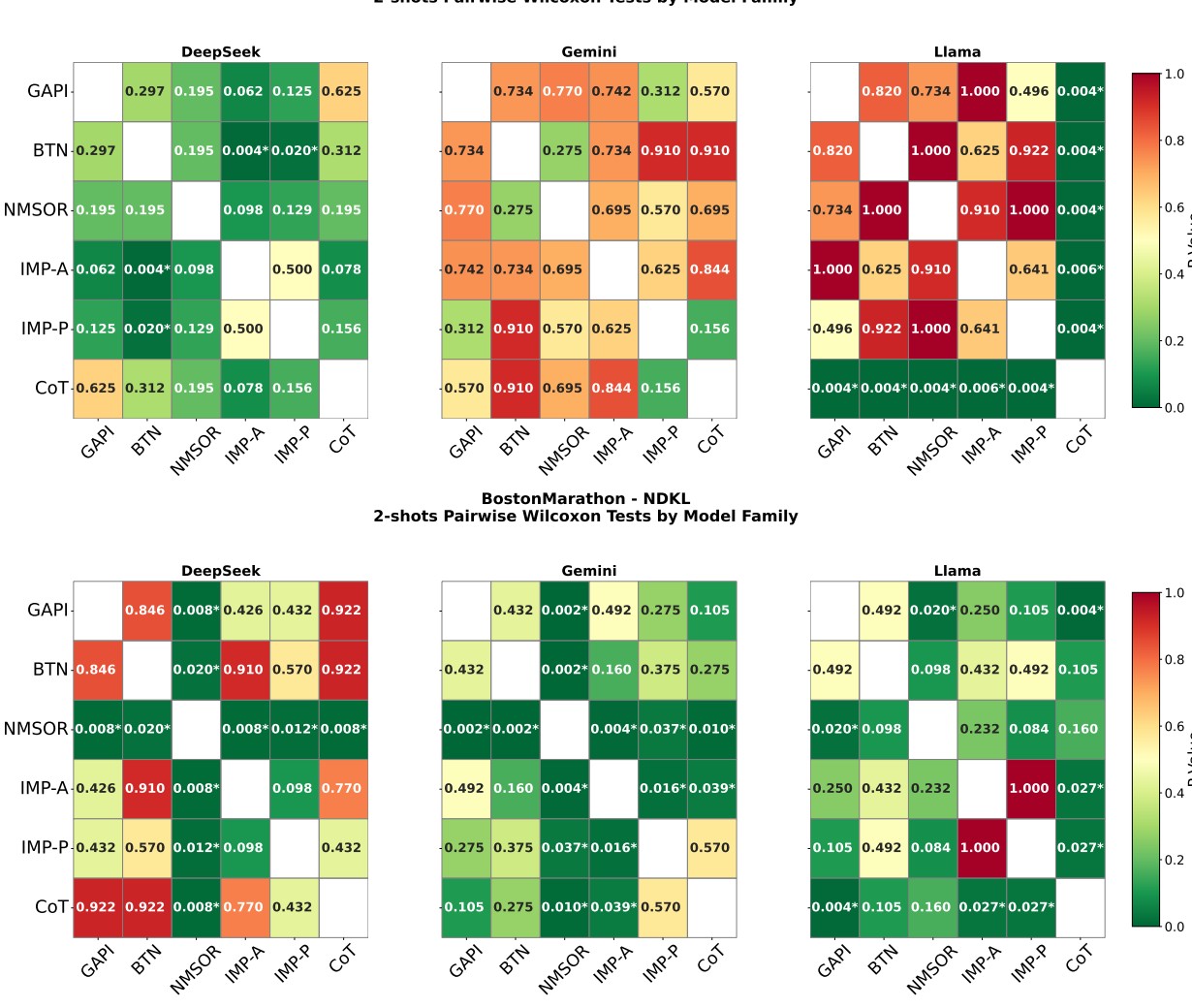

Figure 11: Pipeline approach-vs-approach pairwise Wilcoxon signed-rank tests on the BostonMarathon dataset at the 2-shot configuration. The top row reports comparisons on NDCG@5 (utility) and the bottom row on NDKL (fairness), with each panel showing one LLM family (DeepSeek, Gemini, Llama). Each cell gives the p-value for the comparison between the row method and the column method; asterisks (*) mark significance at $\alpha = 0.05$, and diagonal cells are blank as self-comparisons are undefined. Matrices are symmetric by construction.

**Explicit vs. Implicit Use of Demographic Attributes.** A key empirical distinction explored in this work concerns whether fairness improvements require explicit demographic inference. While strategies that rely on accurate demographic inference can mitigate *allocative harms* (exposure disparity), they introduce significant *representation harms* through misgendering and essentialism (for a taxonomy of sociotechnical harms, we refer the reader to Shelby et al. (2023). Their performance degrades sharply when inference error is high. In contrast, our results suggest that certain LLM-based re-ranking strategies may improve fairness outcomes even without explicitly inferring sensitive attributes. This points to a potentially interesting design space, in which fairness-aware behavior appears to emerge, at least in part, from the model's internal reasoning rather than from explicit demographic labeling.

**Prompt-Objective Gap and Transferability of Findings.** A limitation of this work concerns the relationship between the formal fairness objective defined in Section 3.2 and the prompting strategy utilized to elicit it. The optimization in Equation 1 specifies a precise fairness criterion (e.g., NDKL), yet the prompt

instructs the LLM to "incorporate fairness with respect to {attribute}," without providing a precise formal specification of a particular fairness definition. This design choice isolates the effect of pipeline design, rather than focusing on prompt engineering and analyzing the impact of alternate prompt formats, and is, in that sense, consistent with our goal of auditing LLM behavior under generic conditions. However, this non-formalized notion of fairness also means that different LLMs may interpret the fairness instruction differently, such that the fairness-aware behavior observed in our experiments reflects each model's internal response to the general notion of fairness rather than the optimization of the precise formal objective stated in our formal problem definition section. Findings should therefore be interpreted as characterizing how these specific LLMs behave under this general prompt regime, and may not transfer straightforwardly across different LLM backbones, sensitive attributes, prompt variants, or evaluation metrics. Practitioners adapting these results to deployment contexts should account for this prompt-objective gap. Thus, future work would be beneficial to investigate the impact of prompts with more precisely specified notions of fairness definitions to assess whether tighter alignment between the prompt and a formal fairness criterion yields more transferable outcomes across LLMs.

**Procedural and Distributive Fairness Considerations.** Our evaluation focuses on distributive fairness outcomes, measured through established group-based metrics. The strategies in our study that avoid explicit demographic inference can circumvent mislabeling risks. Yet, they raise procedural fairness questions regarding how fairness is achieved and what information is used in the decision-making process. We do not claim to resolve these normative questions. Instead, we view our results as providing empirical evidence that can inform broader debates about procedural fairness, transparency, and accountability in ranking systems.

**Legal, Ethical, and Governance Implications.** The ability to improve fairness without explicitly inferring sensitive attributes has important legal and ethical implications. In many real-world settings, the collection or inference of demographic attributes is constrained by privacy regulations or anti-discrimination law. Our findings suggest that LLM-based re-ranking may, in some cases, enable fairness interventions that operate under such constraints. However, this possibility also underscores the need for careful governance, as LLMs may implicitly rely on demographic signals in ways that are difficult to audit or explain. Human oversight and institutional safeguards remain essential.

**Scope and Limitations.** Our contribution is intentionally scoped to an empirical investigation of fairness-aware re-ranking under demographic uncertainty. We do not claim algorithmic optimality, nor do we advocate for the deployment of LLM-based ranking systems in high-stakes settings without meaningful human involvement and accountability. We do not attempt to resolve broader societal questions concerning the legitimacy of demographic inference, the role of human judgment, or long-term social impacts. Addressing these issues requires interdisciplinary engagement beyond the scope of this work and represents an important direction for future research.

# 8 Conclusion

This paper examined the possibilities and risks of using LLMs as backbone technologies for fairness-aware re-ranking in settings where sensitive attributes may be missing or uncertain. By evaluating multiple LLM-based re-ranking strategies, we showed that LLMs can support fairness improvements under demographic uncertainty, both when demographic attributes are inferred and, in some cases, even when explicit inference is avoided altogether.

## Ethical Considerations Statement

**Privacy and Ethical Considerations of Demographic Inference.** While demographic attributes are often unavailable due to privacy protections or legal constraints, we acknowledge that the ability of LLMs to infer such attributes raises the very same concerns. Our results should not be interpreted as endorsing demographic inference; rather, the fact that LLMs can accurately infer undisclosed attributes highlights a significant privacy risk, particularly when individuals have intentionally chosen not to disclose sensitive

information. Accordingly, our findings motivate caution and underscore the importance of fairness-aware ranking approaches that avoid explicit demographic labeling, alongside stronger governance and safeguards against misuse.

**Utilization and Access to Tools, Models, and Data Sets.** All components used in this study—including datasets, AI models, and inference tools—were obtained from publicly accessible sources. Our use of these tools was limited to controlled experimental analysis for research purposes only and does not constitute deployment, profiling, or decision-making about real individuals. While some of the experiments in this paper examine the ability of language models or third-party services to infer demographic attributes from names, this inference is studied as an object of analysis rather than as a recommended practice. We do not endorse the use of demographic inference in operational systems, without appropriate legal basis, consent, or institutional safeguards.

Importantly, a central contribution of this work is to demonstrate that fairness-aware re-ranking can, in some cases, be achieved without explicitly inferring or conditioning on sensitive attributes. This finding directly responds to legal, ethical, and policy constraints that restrict demographic inference and motivates alternative fairness interventions that operate under such constraints.

For reference, the licensing terms for the inference tools can be found at the following URLs: **Gen-derAPI**, gender-api.com/en/terms-of-use; **Behind The Name**, behindthename.com/info; **Namsor**: namsor.app/terms-and-conditions. All source code we produced and experimental artifacts of our experimental study will be made available on *github*.

**Broader Impact Statement.** While our research investigates whether LLMs can fairly rank candidates when sensitive attributes are missing, we emphasize that relying on inferred demographic attributes (implicitly or explicitly) should be approached with significant caution. Although some of the methods may improve fairness metrics, they can introduce certain ethical, legal, and governance risks. In particular, the ability of LLM models to recover sensitive attributes from proxies such as names may enable non-consensual inference of demographic information. Such use could undermine principles of transparency, informed consent, and accountability, especially in high-stakes domains such as hiring, lending, education, or admissions. We further note that implicit inference pipelines may be more difficult to monitor and audit than systems using explicitly collected demographic data. Practitioners should therefore not interpret our findings as an endorsement of bypassing consent or regulatory requirements through inferred attributes. Rather, the study highlights the need for careful governance of fairness interventions under missing-data conditions.

Fairness is highly contextual and depends on the task and deployment setting. Any practical use of these methods should therefore be accompanied by careful evaluation, regular auditing, and clear accountability measures, while paying attention to legal and policy implications.

**Researcher Positionality.** Our backgrounds in computer and data science fundamentally shape the lens through which we investigate research questions and design or evaluate computational methods addressing fairness challenges. This perspective informs how we critically engage with the societal implications arising from the integration of digital technologies. We do acknowledge that social scientists and perspectives by other researchers on these critical societal issues is imperative to ensure that potential harm to individuals in our society is avoided to the greatest extend possible.

## Generative AI Usage Statement

No Generative AI tools were used in the creation of the figures or text in this work. AI-based tools were employed only for subsequent light grammatical proof-reading of this manuscript. The conceptual development and intellectual contributions of this work are solely those of the authors, and were conducted independently of LLMs.

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

## A  Appendix

### A.1  Preliminary Experiment: Impact of Explicit Fairness Definitions in Prompt Instructions.

In Section 4.1.2, we work with a simple fairness instruction. To examine whether more explicitly defining the notion of fairness would change model behavior, we first asked the models to describe two fairness notions commonly studied in the literature: representational fairness (maintaining group presence in line with a target distribution) Geyik et al. (2019) and exposure fairness (ensuring groups receive comparable visibility through ranking positions) Singh and Joachims (2018). We then incorporated these definitions into the prompts and compared the resulting rankings with those produced by the basic fairness instruction used in our main experiments.

We find that, once the sensitive attribute is already specified, explicitly adding these fairness definitions leads to only minor differences in fairness outcomes relative to the baseline prompt. Table 5 presents the definition of the fairness notions generated by the LLMs, while Table 4 reports the corresponding NDKL values obtained with the Gemini model under each prompt variant. These results are similar to the ones reported by Olulana et al. (2026).

| Fairness Instruction | NDKL Value |
|---|---|
| Initial Ranking | $0.21 \pm 0.07$ |
| BASE | $0.06 \pm 0.00$ |
| Representational | $0.08 \pm 0.04$ |
| Exposure | $0.10 \pm 0.04$ |

Table 4: Comparison of fairness instructions using NDKL (lower is better).

| Fairness Notion | Verbatim Definition |
|---|---|
| **Fairness with respect to sex** | `Fairness with respect to sex is a complex concept because it is interpreted differently depending on the context--legal, social, biological, or economic.  Generally, it revolves around the tension between equality of opportunity (treating everyone the same) and equity (recognizing different needs to achieve fair outcomes).` |
| **Fairness in representation with respect to sex** | `Fairness in representation with respect to sex refers to the principle that individuals of all sexes and genders should be portrayed, included, and empowered in a manner that is equitable, accurate, and free from bias.` |
| **Fairness in exposure with respect to sex** | `In the context of machine learning and algorithmic systems, fairness in exposure (often called exposure fairness) refers to the principle that different groups--in this case, defined by sex--should receive equitable visibility, attention, or opportunity within a system's output.` |

Table 5: Verbatim LLM-generated (Gemini) definitions of alternative fairness notions with respect to sex. Monospaced formatting is used to indicate that the text is reproduced verbatim.

## A.2 Evaluation Using an Alternative Fairness Metric: Average Exposure Ratio

### Average Exposure Ratio (AveExpR CI-95) (Singh and Joachims (2019))

. The Average Exposure Ratio quantifies fairness in terms of visibility by comparing the average exposure allocated to the disadvantaged group against that received by the advantaged group in a given ranking $\tau$:

$$Average\ Exposure\ Ratio(\tau) = \frac{Average\ Exposure(\tau, dis)}{Average\ Exposure(\tau, adv)}, \tag{7}$$

Here, the average exposure for any group $g$ is computed as:

$$Average\ Exposure(\tau, g) = \frac{1}{|a|} \sum_{a_i \in g} Exposure(\tau, a_i),$$

where individual exposure is determined by position-based visibility:

$$Exposure(\tau, a_i) = \frac{1}{\log_2(\tau(a_i) + 1)}.$$

In our evaluation, we average this ratio across 10 independent runs (that is, 10 individual test sets) and report 95% confidence intervals. A ratio of 1.0 indicates equitable exposure, with values below or above signifying under- or over-exposure of the disadvantaged group, respectively.

As an additional evaluation, we report results using the fairness metric AveExpR, defined in Section 5.1. Since this metric measures the ratio between the average exposure of disadvantaged and advantaged groups, it is applicable only to datasets with binary sensitive attributes. Therefore, we report AveExpR results only for those datasets. Figure 12 presents the average exposure ratios for the BOSTON MARATHON dataset, while results for the (W)NBA and COMPAS datasets are shown in Figures 14 and 15, respectively.

Overall, the average exposure results are consistent with the NDKL findings in addressing our research questions, as discussed in Section 6. The Initial ranking exhibits lower exposure ratios, indicating imbalance between the groups. Applying fairness-aware strategies substantially improves AveExpR. For Gemini and DeepSeek, most methods bring AveExpR close to the ideal value of 1.0, with several configurations ranging between approximately 0.95 and 0.99, particularly when using inference methods with higher accuracy. We also observe that providing shots tends to move the results closer to the fair re-ranking baseline, DCS, especially when the inferred sensitive attributes are more accurate.

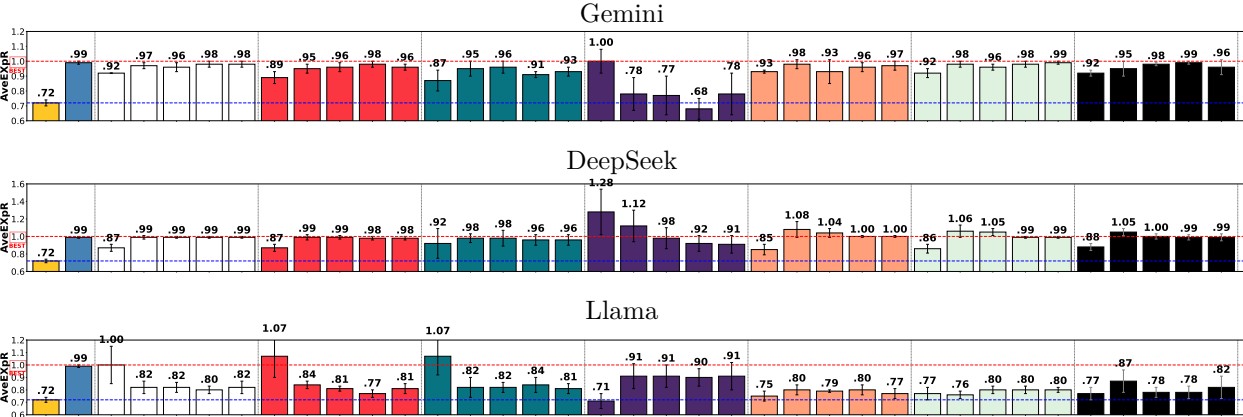

Figure 12: Additional Fairness results (AveExpR) across LLM models, Gemini (top), Deepseek (middle), Llama (bottom) on BOSTON MARATHON

## A.3 Terminology: Performance vs. Utility

The information retrieval literature frequently refers to the existence of a fairness-utility trade-off, a standard formulation where fairness is modeled as a constraint that penalizes the utility function (e.g., measured in

terms of NDCG). However, this framing has been actively contested by authors such as Harned and Wallach (2019) and Weerts et al. (2024), who argue that framing fairness as a trade-off against utility implies that the current performance represents an objective or neutral "gold standard". If the historical data is biased, (e.g., hiring decisions that excluded women), then a model that reproduces this data has high "accuracy" but low "truth". In this view, fairness interventions do not degrade utility; they correct for measurement error in the ground truth. Rather than referring to a fairness-utility trade-off, it would be more correct to a tension between fairness constraints—as enforcement of distributional equity— and observed performance— as retention of historical relevance signals.

## A.4  Terminology: Gender vs. Sex

Recent scholarship including the influential work of Devinney et al. (2022) on theories of gender in NLP and Lockhart et al. (2023) on name-based inference, emphasizes a rigorous distinction between biological sex and social gender. After all, names are social signals that connote gender presentation or cultural norms; they do not encode biological sex. While it would be more correct to state that the demographic inference tools used in this paper infer *gender*, we deliberately opt to use the term *sex* as these tools only output 'male' and 'female' values—demographic axes that are mostly associated with biological sex at birth. Moreover, the use of sports-related datasets—namely, Boston Marathon and (W)NBA—entails the use of sex as an administrative category that determines eligibility. Our study here measures the alignment between social signaling (names) and administrative categorization (competition sex categories). It is important to note that this alignment may not hold for transgender athletes or individuals whose gender identity diverges from their administrative sex category. The authors do not endorse any form of biological essentialism.

## A.5  List Size Selection and Design Rationale

Our task is fair re-ranking rather than full ranking. We assume an upstream system produces an initial utility-ranked list over a large candidate pool, and that fairness-aware re-ranking is applied only to the decision-relevant top-k portion, as is common in real-world deployments (e.g., hiring shortlists, scholarship finalists, interview callbacks). Accordingly, all methods in this paper operate on a fixed top-k list.

Although modern LLMs support increasingly large context windows, token limits and output stability remain practical constraints, particularly when fairness objectives require structured reasoning over many candidates. To guide our experimental design, we analyze how fairness outcomes vary with list size.

Figure 13 reports fairness metrics for different values of $k$ when using DeepSeek as the backbone model. We observe that fairness improves consistently relative to the initial ranking across list sizes, but that $k = 50$ yields the most stable trade-off between representation fairness (NDKL) and exposure balance (AveExpR). While larger lists occasionally achieve comparable average exposure, they exhibit higher variance in fairness outcomes. Based on this stability criterion, we select $k = 50$ for gender experiments. For race-based experiments, we use $k = 100$ to account for greater sparsity and uncertainty in race inference, while maintaining a comparable decision-relevant scope. All methods are evaluated under the same top-k constraint for a given experiment.

We emphasize that this analysis is not intended to claim generalizability to full-list ranking. Instead, it motivates a controlled and realistic setting in which to evaluate whether LLM-based fair re-ranking is feasible at all under demographic uncertainty. Extending these approaches to longer lists and full-ranking scenarios remains an important direction for future work.

## A.6  Full Results on (W)NBA Data Set

Figure 14 show the results across 0-4 shots across all models on the NBA/WNBA dataset. The results are consistent with those discussed in Section 6.4.

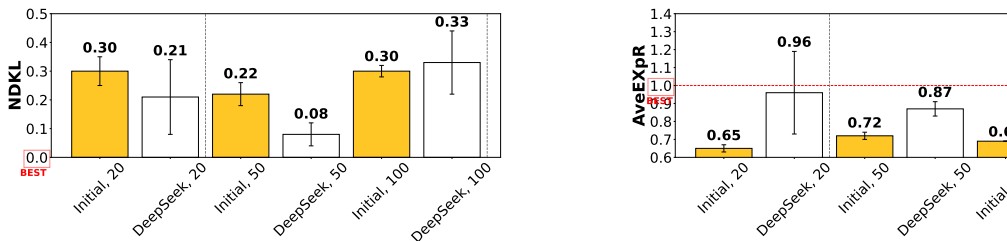

Figure 13: Initial Fairness (AveExpR, NDKL) for DeepSeek across different rank sizes on BOSTONMARATHON Data Set.

Figure 14: Fair ReRanking results across LLM models, Gemini (top), Deepseek (middle), Llama (bottom) on W(NBA), evaluated using AveExpR, NDKL, and NDCG@5 metrics.

## A.7   Full Results on COMPAS Dataset

Figure 15 show the results across 0-4 shots across all models on the COMPAS dataset. The results are consistent with those discussed in Section 6.4.

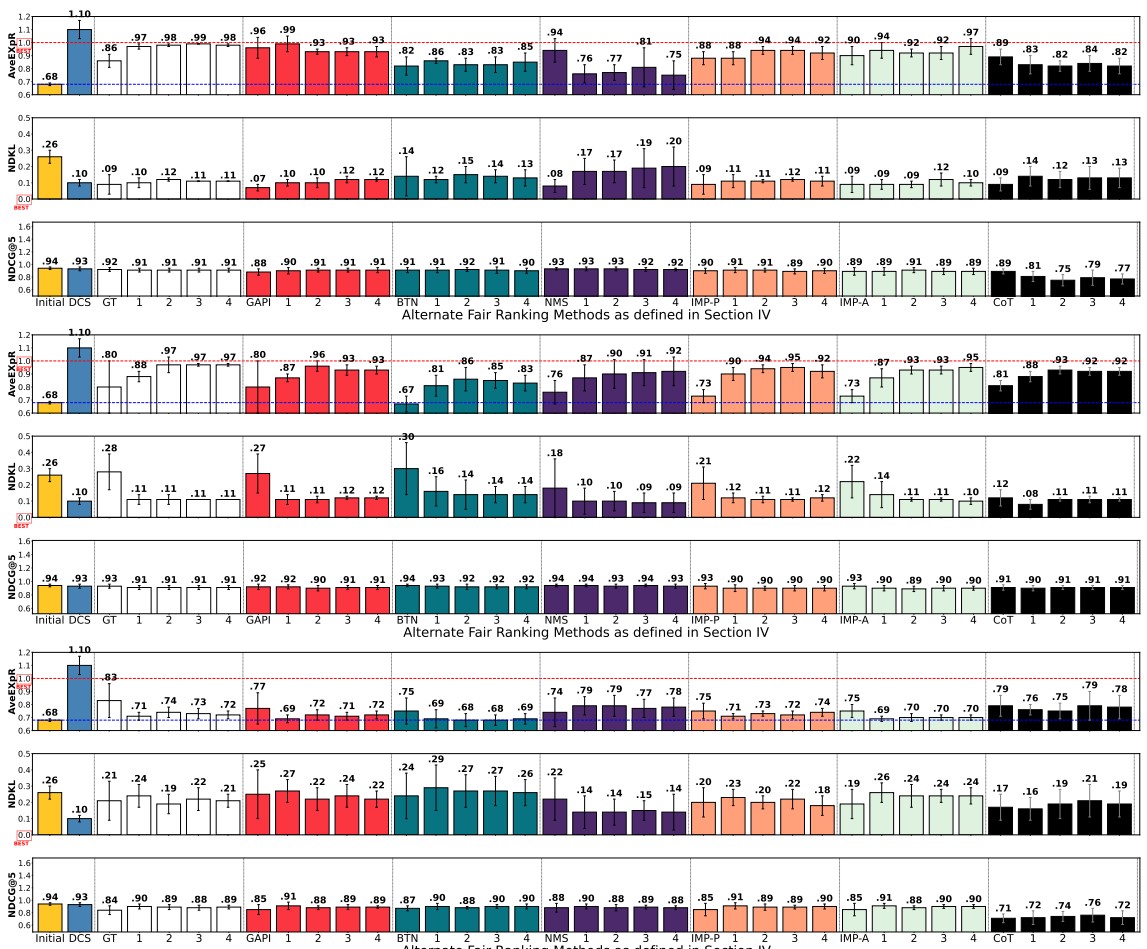

Figure 15: Fair ReRanking results across LLM models, Gemini (top), Deepseek (middle), Llama (bottom) on COMPAS, evaluated using AveExpR, NDKL, and NDCG@5 metrics.

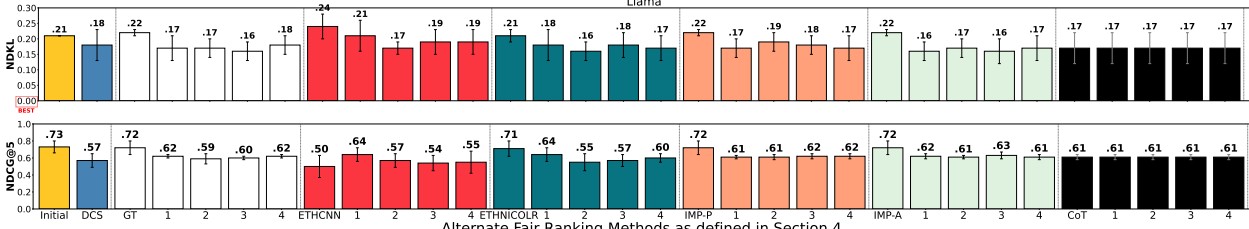

Figure 16: Fair ReRanking results on Llama for STARTUP FOUNDERS, evaluated using AveExpR, NDKL, and NDCG@5 metrics.

## A.8  Full Results on Startup Founders Dataset

Figure 16 show the results across 0-4 shots across for Llama model on the Startup Founders dataset. The results are consistent with those discussed in Section 6.4.

## A.9  Sample prompts

Figures 17, 18 and 19 show the prompts for EXP, IMP and CoT-FairRank as discussed in Section 4 of the main paper.

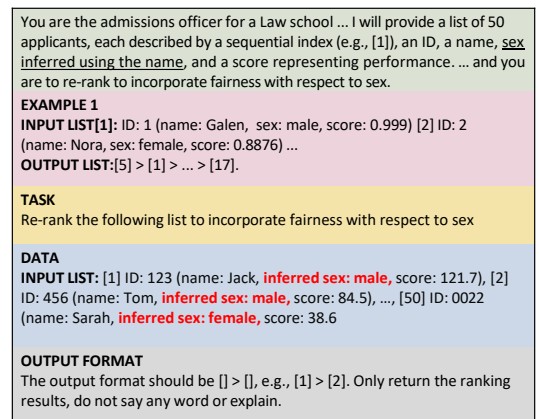

Figure 17: Approach 1: Explicit Inference via Third-Party Services, a 1-shot Example Prompt

Figure 18: Approach 2: LLM-Based Fair Re-ranking with Implicit Inferencing (1-shot). Implicit-A: complete prompt without sex attribute in shots.

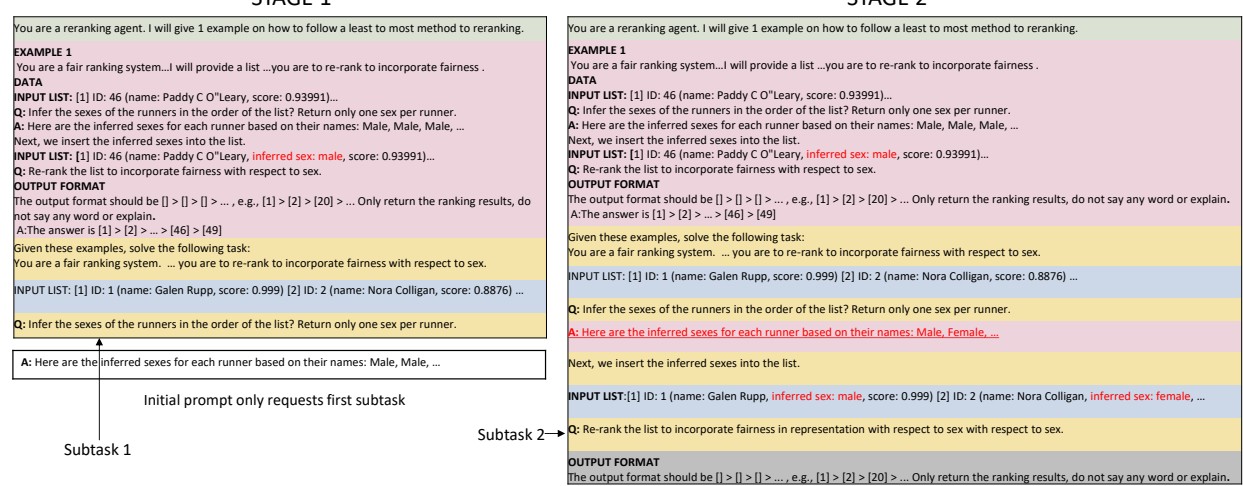

Figure 19: Approach 3: Chain-of-Thought Fair Re-ranking strategy prompting with one-shot examples.

## A.10 Fairness Results for Approach 2 across LLMs

Figures 20 and 21 show fairness results for Approach 2 as discussed in Section 6.5 in the main paper.

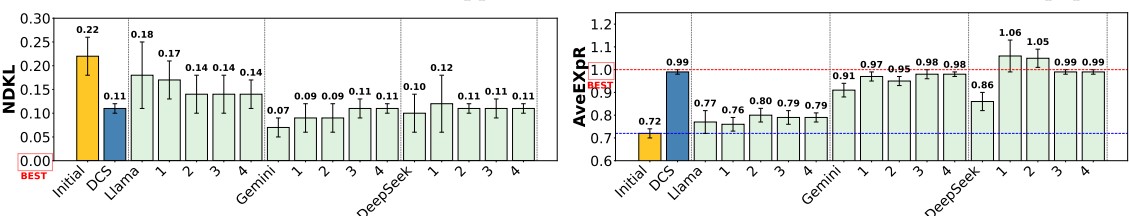

Figure 20: Fairness (AveExpR, NDKL) for IMP-A across 3 LLM model types with shots varying from 0 to 5 on BostonMarathon Data Set.

| Dataset | Primary Withheld Attributes | Additional Holdouts (if needed) |
|---|---|---|
| COMPAS | sex, race | age, priors_count, juv_fel_count, juv_misd_count, juv_other_count, c_charge_degree, decile_score, two_year_recid |
| Boston Marathon | Gender, Age | City, State, Country, Citizen, Pace, Proj Time, score, Overall, Division |
| Startup Founders | gender, perceived_race_majority | ethnicolor, ethcnn, Inflation adjusted Amount, ProfileURL |
| NBA / WNBA | Gender | NumSeasons, AvgPER, CareerPoints |

Table 6: Attributes withheld per row in MELD. At least two attributes are withheld for each row to prevent the task from reducing to single-attribute inference. When fewer than two sensitive attributes are available, additional non-sensitive fields are automatically withheld.

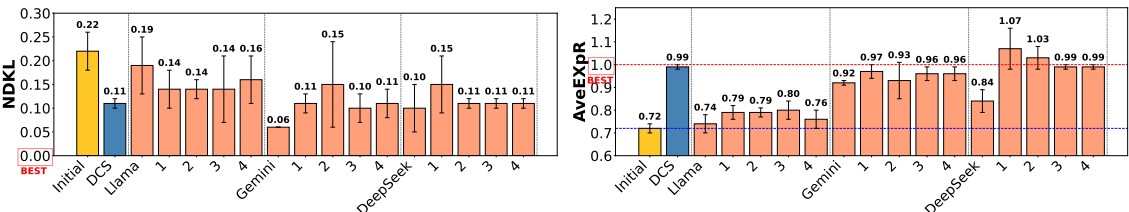

Figure 21: Fairness (AveExpR, NDKL) for IMP-P across 3 LLM model types with shots varying from 0 to 4 on BOSTONMARATHON Data Set.

## A.11 MELD: Memorization Effects Levenshtein Detector

To assess whether demographic inference results could be explained by memorization rather than generalization, we employ MELD (Memorization Effects Levenshtein Detector) (Nori et al. (2023)), a lightweight string-similarity diagnostic. MELD measures normalized Levenshtein distance between model outputs and held-out ground-truth values, flagging near-exact matches (similarity $\geq 0.95$) as potential memorization. In our study, MELD is applied in a tabular completion setting, where multiple attributes (e.g., sex and race) are withheld from each record and the model is prompted to reconstruct them from remaining fields. This analysis serves as a conservative check for verbatim recall and supports the conclusion that observed inference behavior is not solely attributable to memorization.

**Procedure.** For each dataset, MELD withholds at least two attributes per row (See Table 6) to prevent memorization detection from degenerating into single-attribute inference. Sensitive attributes are prioritized when available (e.g., sex, race). If fewer than two such attributes are present, additional non-sensitive fields are automatically withheld from the same row. The LLM is prompted to reconstruct all withheld fields jointly, and near-exact reconstruction is treated as evidence of memorization.

**Memorization Test Results.** Table 7 reports the MELD memorization results across all datasets and LLM backbones. For Gemini, DeepSeek, and LLaMA, we observe zero near-exact matches under a 95% similarity threshold on all datasets, providing no evidence of record-level memorization of withheld demographic attributes. While no near-exact matches are observed for any model or dataset under the 95% similarity threshold, the mean Levenshtein ratios vary across models and datasets, reflecting differences in how closely generated outputs resemble the withheld attribute values. Lower ratios indicate that the generated text differs substantially from the ground truth, suggesting little overlap at the string level, whereas higher ratios reflect partial similarity without approaching verbatim reconstruction.

Across all four datasets—COMPAS, Boston Marathon, Startup Founders, and NBA/WNBA—the MELD analysis finds no evidence of record-level memorization by any evaluated LLM. For every dataset, the memorization percentage is 0%, with zero near-exact matches under the 95% Levenshtein similarity threshold.

However, the mean Levenshtein ratios differ across datasets, reflecting differences in dataset structure and attribute regularity rather than memorization.

| Dataset | Model | Samples | Memorization (%) | Mean Lev. Ratio |
|---------|-------|---------|------------------|-----------------|
| COMPAS | DeepSeek-V3 | 50 | 0.00 | 0.3519 |
| Boston Marathon | DeepSeek-V3 | 50 | 0.00 | 0.5556 |
| Startup Founders | DeepSeek-V3 | 50 | 0.00 | 0.3599 |
| NBA/WNBA | DeepSeek-V3 | 50 | 0.00 | 0.4603 |
| COMPAS | LLaMA-3-8B-Instruct (local) | 50 | 0.00 | 0.1575 |
| Boston Marathon | LLaMA-3-8B-Instruct (local) | 50 | 0.00 | 0.1689 |
| Startup Founders | LLaMA-3-8B-Instruct (local) | 50 | 0.00 | 0.1296 |
| NBA/WNBA | LLaMA-3-8B-Instruct (local) | 50 | 0.00 | 0.1844 |
| COMPAS | Gemini | 50 | 0.00 | 0.0000 |
| Boston Marathon | Gemini | 50 | 0.00 | 0.0000 |
| Startup Founders | Gemini | 50 | 0.00 | 0.0000 |
| NBA/WNBA | Gemini | 50 | 0.00 | 0.0000 |

Table 7: MELD memorization analysis across datasets and LLM backbones using a 95% Levenshtein similarity threshold. No near-exact matches were observed for any model or dataset, indicating no evidence of training-data memorization under the MELD criterion.

### A.12 Preliminary Tests: Varying Group Percentages

We acknowledge that real-world group distributions are often imbalanced, and therefore examine performance under varying group proportions to assess how the method behaves beyond balanced settings. Figure 22 shows how varying the proportion of the disadvantaged group affects ranking fairness. We compare the utility-based ranking (yellow line) with fairness-aware rankings (blue line) produced by Approach 1 with ground truth demographic labels. We use this method as a base method to check how an LLM would fairly re-rank with different group balance. Across most group proportions, the fairly re-ranked results achieved NDKLs lower than the Initial ranking, indicating improved fairness under both balanced (50%) and imbalanced settings. At more extreme proportions (80%–90%), fairness becomes more challenging for both methods, and the gap between the approaches narrows. Nevertheless, the reranked outputs remain competitive with, and in some cases superior to, the Initial ranking.

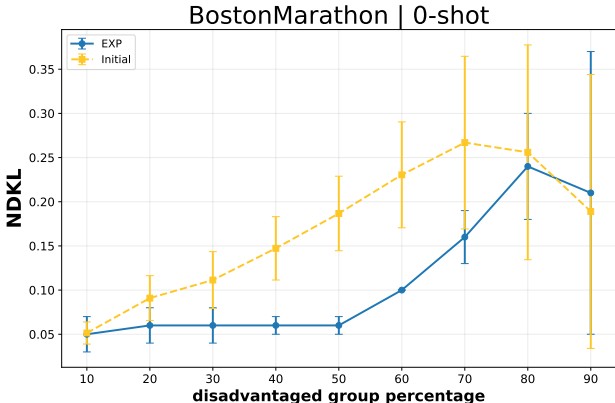

Figure 22: Effect of disadvantaged-group proportion on ranking fairness (NDKL) in the Boston Marathon dataset under the 0-shot setting using Gemini.

### A.13 Prompts

---

**Approach 1**

You are a fair ranking system. You are the ranking officer for a marathon and your goal is to rank runners by predicted success in the marathon, while ensuring fairness. I will provide a list of 50 runners, each described by a sequential index (e.g., [1]), an ID, a name, sex inferred using the name, and a score representing performance. The list is already ranked by official time scores, from fastest to slowest, and you are to re-rank to incorporate fairness with respect to sex.

TASK
Re-rank the following list to incorporate fairness with respect to sex.

DATA
INPUT LIST:  [1] ID: 1 (sex: male, score: 0.99756), [2] ID: 2 (sex: male, score: 0.89783), ..., [50] ID: 50 (sex: female, score: 0.44591)

OUTPUT FORMAT
The output format should be [] > [] > [] > ... , e.g., [1] > [2] > [20] > ... Only return the ranking results, do not say any word or explain.

---

**Approach 2**

You are the ranking officer for a marathon and your goal is to rank runners by predicted success in the marathon, while ensuring fairness. I will provide a list of 50 runners, each described by a sequential index (e.g., [1]), an ID, a name and a score representing performance. The list is already ranked by official time scores, from fastest to slowest, and you are to re-rank to incorporate fairness with respect to sex.

TASK
Re-rank the following list to incorporate fairness with respect to sex.

DATA
INPUT LIST:  [1] ID: 1 (name: Galen Rupp, score: 0.999) [2] ID: 2 (name: Nora Colligan, score: 0.8876), ..., [50] ID: 50 (name: Toshikazu Shida, score: 0.40297).

OUTPUT FORMAT
The output format should be [] > [] > [] > ... , e.g., [1] > [2] > [20] > ... Only return the ranking results, do not say any word or explain.

**Approach 3: Stage 1**

You are a fair ranking system. You are the ranking officer for a marathon and your goal is to rank runners by predicted success in the marathon, while ensuring fairness. I will provide a list of 50 runners, each described by a sequential index (e.g., [1]), an ID, a name, sex inferred using the name, and a score representing performance. The list is already ranked by official time scores, from fastest to slowest, and you are to re-rank to incorporate fairness with respect to sex.

INPUT LIST:  [1] ID: 1 (name: Galen Rupp, score: 0.999) [2] ID: 2 (name: Nora Colligan, score: 0.8876), ..., [50] ID: 50 (name: Toshikazu Shida, score: 0.40297).

Q: Infer the sexes of the runners in the order of the list? Return only one sex per runner.

**Approach 3: Stage 2**

You are a fair ranking system. You are the ranking officer for a marathon and your goal is to rank runners by predicted success in the marathon, while ensuring fairness. I will provide a list of 50 runners, each described by a sequential index (e.g., [1]), an ID, a name and a score representing performance. The list is already ranked by official time scores, from fastest to slowest, and you are to re-rank to incorporate fairness with respect to sex.

INPUT LIST:  [1] ID: 1 (name: Galen Rupp, score: 0.999) [2] ID: 2 (name: Nora Colligan, score: 0.8876), ..., [50] ID: 50 (name: Toshikazu Shida, score: 0.40297).

Q: Infer the sexes of the runners in the order of the list? Return only one sex per runner.

A: {Model's inferred is result inserted here...}

Next, we insert the inferred sexes back into the list.

INPUT LIST:  [1] ID: 1 (name: Galen Rupp, inferred sex: male, score: 0.999) [2] ID: 2 (name: Nora Colligan, inferred sex: female, score: 0.8876), ..., [50] ID: 50 (name: Toshikazu Shida, inferred sex: male, score: 0.40297)

Q: Re-rank the list to incorporate fairness in representation with respect to sex with respect to sex.

OUTPUT FORMAT
The output format should be [] > [] > [] > ... , e.g., [1] > [2] > [20] > ... Only return the ranking results, do not say any word or explain.

