# OpenReview forum: "Can LLMs Rank Candidates with Missing Sensitive Attributes Fairly?"
_TMLR — Accepted by TMLR_

### Review · Reviewer_jy3K · 2026-04-03

**Summary Of Contributions:**

This work studies how LLM-based fair re-ranking is impacted when demographic information is missing in the data. It sets three approaches to conduct the experiments: 1) presence of sensitive attributes using traditional third-party services; 2) absence of sensitive attributes; 3) chain-of-thought prompting to encourage LLM infer sensitive attributes. Main findings from the experiments include that LLMs match the accuracy of leading third-party services in demographic inference, and LLMs is capable to embed fairness objectives into rankings even without explicitly inferring sensitive attributes.

**Strengths**
1. Clear research questions and experimental design. The paper present clear strategies to study the problems from multiple dimensions. The structure of the experiments is strong.
2. Extensive comparative experiments: The paper compares several third-party inference tools, multiple LLM backbones and datasets and analyzes their performances.

**Weaknesses**
1. Unclear role of Equation (1) in the actual experiments. The paper formalizes fair re-ranking through Equation (1), where utility and fairness are combined via trade-off coefficients $\alpha$ and $\beta$. However, the paper does not explain how these parameters are chosen in the experiments. The connection between the formal objective and the implemented system is unclear.
2. Lack of justification and sensitivity analysis for the utility–fairness trade-off. Equation (1) assumes that re-ranking quality can be captured by a weighted combination of utility and fairness, but the paper does not justify why this scalarized objective is the appropriate formulation for the studied setting. Also, the paper does not analyze how varying these weights would affect the results. Without such discussion or ablation, it is difficult to know whether the reported conclusions are robust across different fairness–utility preferences.
3. Writing and result figures are often unclear and imprecise. The paper contains multiple passages where the description is difficult to follow. Figures are poorly presented and hard to interpret. (See the detailed comments below)
4. Insufficient dataset statistics. The paper does not provide enough basic descriptive statistics for the datasets, such as dataset sizes, group distributions, or other characteristics needed to understand the benchmark setting.

**Audience:**

Yes

**Audience Explanation:**

This paper is about the application of LLMs.

**Claims And Evidence:**

Yes

**Claims Explanation:**

The paper conducts extensive experiments and analyzes the results.

**Requested Changes:**

Detailed comments regarding writing and presentation issues:
1. In the first paragraph of section 3.2, the definiton of 'utility' and 'fariness objectives' should be clearly defined there, as they are foundational concepts for later parts, particularly helpful for readers who are unfamiliar with those concepts.
2. In Equation (1), the definition of $s$ should be defined clearly.
3. Last row of section 4.1.1, missing period before 'For example'.
4. In section 5 Dataset, it would be good to describe some statistics of the datasets, and how the ground-truth scores are calculated for each dataset. Also, a compile error in (Appendix ??).
5. In section 5.1 Evaluation Metrics, the metric 'Average Exposure Ratio' seems to be a missed item.
6. The title of Figure 3 has formatting issue.
7. Lack description of the meaning of different colors in the figures.
8. Title of Figure4: there is the results of Boston Marathon dataset but it is missing in the title.

---

> ### Author Response · Authors · 2026-04-21
> **Clarifications, Presentation, and Dataset Statistics**
>
> __*The revised manuscript will be uploaded after all comments have been addressed, with changes related to this reviewer highlighted in green ink.*__
>
> We thank the reviewer for seeing our experimental design as clear, and that our comparative evaluation across models, tools, and datasets has breadth.
> We appreciate your feedback highlighting areas that would benefit from further clarification; and address all your suggestions in the revised document, as we explain below.
>
> __W1 \& W2. Clarify role of Equation (1) in experiments, in particular, concerning the utility-fairness trade-off.__
>
> Equation (1) was meant to serve as a conceptual formulation of the fairness--utility trade-off of a standard fair re-ranking problem, aligned with prior literature [1]. However, we do not aim to explicitly optimize this scalarized trade-off objective in this paper; rather, the LLM is guided through prompting to produce fairness-aware rankings in settings where demographic attributes are unavailable, while we systematically vary the intervention strategies used to handle these missing attributes. The generated ranking and its properties are then evaluated with respect to utility and fairness metrics.
>
> In the few shot setting, exemplars are constructed using the DetConstSort algorithm, where the task is better represented as a utility optimization with fairness constraints. We agree that the explicit listing of trade-off parameters $\alpha$ and $\beta$ may set the wrong expectation for the reader. For clarity, we thus revise this formulation into a constrained view notion, removing explicit references to the trade-off parameters $\alpha$ and $\beta$. _(See Section 3.2 in revision)_
>
> __Old equation:__
>
> $$
> \tau^{*}=\arg\max_{\tau\in S_n}\left[\alpha\,U(\tau;s)-\beta\,\mathcal{L}_{\mathrm{fair}}(\tau;a)\right]
> $$
>
> __Revised equation:__
>
> $$
> \tau^{*}=\arg\max_{\tau\in S_n} U(\tau;s)
> \quad \text{s.t.} \quad
> \mathcal{L}_{\mathrm{fair}}(\tau;a)\leq \epsilon
> $$
>
> ---
>
> __W3. Improve figure and writing clarity, based on the detailed suggestions provided below.__
>
> We thank you for your suggestions to help us improve the clarity of our writing and presentation of our figures. We have addressed these suggestions in the revised manuscript in the specific locations indicated below _(interleaved after each writing suggestion)._
>
> 1.  __In the first paragraph of section 3.2, the definition of 'utility' and 'fairness objectives' should be clearly defined there, as they are foundational concepts for later parts, particularly helpful for readers who are unfamiliar with those concepts.__
>
>  Addressed in 3.2 (as written below).
>
> _Here, utility refers to the quality of the ranking with respect to observed relevance scores, typically measured using metrics such as NDCG. Fairness objectives refer to constraints or criteria that regulate how exposure or representation is distributed across groups defined by the sensitive attribute._
>
> 2. __In Equation (1), the definition of $s$ should be defined clearly.__
>
>  Addressed in 3.2.
>
> _score vector $s = (s_1, . . . , s_n)$_
>
> 3. __Last row of section 4.1.1, missing period before 'For example'.__
>
> Period added.
>
> 4. __In section 5 Dataset, it would be good to describe some statistics of the datasets, and how the ground-truth scores are calculated for each dataset. Also, a compile error in (Appendix ??).__
>
> More details about datasets will be added _(Including group percentages)_. We fixed the compile error for the Appendix in revision.
>
> 5. __In section 5.1 Evaluation Metrics, the metric 'Average Exposure Ratio' seems to be a missed item.__
>
> In revision, we moved the definition to the Appendix, where it is used.
>
> 6. __The title of Figure 3 has formatting issue.__
>
> Extra description removed in revision.
>
> 7. __Lack description of the meaning of different colors in the figures.__
>
>  Added in Figure 3 in revision.
>
> 8. __Title of Figure 4: there is results of Boston Marathon dataset but it is missing in the title.__
>
> Title added in revision.
>
> ---
>
> __W4: Insufficient dataset statistics.__ The paper does not provide enough basic descriptive statistics for the datasets, such as dataset sizes, group distributions, or other characteristics needed to understand the benchmark setting.
>
>
> We will add more information about the group distributions for each data set in Section 5 _(the data set description section)_ in the revised manuscript. This augments the existing information provided in the ``Test set preparation'' subsection, where we explain the composition of specific ranking sets that are extracted out of these overall larger data sets and sent to the LLMs _(i.e., subset and group sizes)_. Additionally, we report baseline fairness and utility metrics computed on the ground-truth ranking sets _("Initial" in Section 6)._
>
> ---
>
> __References__
>
> __[1]__ Ashudeep Singh and Thorsten Joachims. 2018. _Fairness of Exposure in Rankings._ Proceedings of the ACM SIGKDD Conference (2018), 2219--2228.

---

> > ### Author Response · Authors · 2026-04-26
> > **Revised manuscript uploaded**
> >
> > **We have uploaded the revised manuscript incorporating the changes discussed in our earlier response.**
> >
> > All revisions made in response to your comments are highlighted in **green** throughout the manuscript for ease of review.
> >
> > We thank the reviewer again for the detailed and feedback.

---

### Review · Reviewer_UTcH · 2026-04-10

**Summary Of Contributions:**

This paper studies fairness-aware LLM reranking when sensitive demographic attributes are missing, which is a practically important setting that is underexplored relative to prior work assuming access to protected attributes. The paper proposes and compares three pipeline designs: reranking with third-party inferred attributes, reranking without explicit inference, and a chain-of-thought pipeline that first infers attributes and then reranks. Empirically, it evaluates these approaches across multiple datasets, models, inference services, and fairness/utility metrics, and reports that LLMs can sometimes improve fairness even without explicit demographic labels, while performance is highly sensitive to inference quality and is further shaped by few-shot demonstrations. I see the main strengths as the timeliness of the problem framing, the comparative experimental setup across pipeline choices, and the paper’s attempt to discuss governance and privacy risks rather than presenting demographic inference as an unqualified solution. The main weaknesses are that the prompt-level notion of “fairness” is intentionally underspecified, several experimental choices reduce real-world realism, and some conclusions would be stronger with more rigorous robustness and statistical analysis.

**Audience:**

Yes

**Audience Explanation:**

I think this paper would be of clear interest to at least part of the TMLR audience, especially researchers working on LLM evaluation, algorithmic fairness, ranking/retrieval, and responsible deployment in high-stakes decision settings. The paper addresses a realistic tension: fairness methods often require protected attributes, but those attributes may be unavailable for legal, practical, or ethical reasons. The finding that LLMs may still improve group-fairness metrics without explicit labels is both interesting and concerning, because it opens a design space for fair intervention while also raising the possibility of implicit protected-attribute inference that is harder to audit. This combination of empirical results and governance implications makes the work relevant beyond narrow benchmarking.

**Broader Impact Concerns:**

The paper already includes an ethical considerations and broader impact discussion, so I do not think a wholly missing Broader Impact Statement is the problem. My concern is instead that the current discussion should more explicitly address misuse pathways. The paper shows that LLMs can often recover protected attributes from names and may improve fairness even without explicit labels, but this same result also creates a risk that practitioners will use “implicit” fairness methods as a way to sidestep consent, transparency, or audit requirements. I would therefore encourage the authors to expand the broader impact discussion around covert profiling, legal and policy compliance, the risk of non-consensual protected-attribute recovery, and the danger that implicit inference may be harder to monitor than explicit pipelines. The paper is aware of these concerns, but the safeguards and deployment implications could be stated more concretely.

**Claims And Evidence:**

No

**Claims Explanation:**

The paper provides a somewhat empirical basis for its central claims. It evaluates three distinct LLM-based pipeline designs, compares them against both initial utility rankings and a traditional fair reranking baseline, and uses standard fairness and utility metrics such as NDKL, exposure ratio, and NDCG. It also compares LLM-based demographic inference against multiple third-party inference services and shows a broad spread in inference quality, which is important for supporting the claim that reranking outcomes are sensitive to inference accuracy. In addition, the authors explicitly scope the work as an empirical audit rather than an optimization claim, and they acknowledge limitations such as excluding “unknown” race predictions and not recommending deployment of demographic inference in practice.

However, the following changes made the claims unreliable. First, the prompt asks models to “incorporate fairness” without instantiating a concrete fairness definition, so it is difficult to know whether the models are learning a stable fairness objective or simply responding to vague alignment cues. Second, the few-shot examples are produced by a classical reranking algorithm, which makes it harder to disentangle genuine reasoning from imitation of the example pattern. Third, some experimental design decisions, such as balanced subset construction and exclusion of unknown race labels, may limit ecological validity. So while the evidence is sufficient for the paper’s bounded empirical claims, some stronger interpretations should be softened or further substantiated.

**Requested Changes:**

Critical:

1. The paper should clarify the operational meaning of “fairness” in the prompting setup. Because the instruction deliberately leaves fairness unspecified, it is currently hard to interpret what exactly the model is optimizing and whether comparisons across methods are conceptually clean. At minimum, the authors should discuss this more explicitly and ideally add a sensitivity analysis with alternative prompt phrasings or more explicit fairness criteria.
2. The paper should strengthen the experimental validation around robustness and realism. In particular, I would like to see either evaluation on more natural group distributions or a stronger justification for the balanced subset construction, since this choice may affect both fairness and utility behavior relative to deployment settings. Relatedly, the exclusion of “unknown” race labels should be discussed more thoroughly in the main paper, because this may remove precisely the difficult cases where demographic uncertainty matters most.
3. The paper would benefit from stronger statistical treatment. The current presentation is suggestive, but the paper should more clearly report variance and statistical significance for the main fairness and utility comparisons, not only for selected metrics. This is particularly important because some improvements appear modest and because the conclusions compare multiple models, prompting settings, and inference pipelines.

Would be helpful:
The scope and limitations section is directionally good, but the paper should connect the empirical findings more directly to deployment guidance. In particular, the manuscript should explain how organizations should interpret the result that implicit reranking may work without explicit attribute labels, given that such behavior can still amount to latent protected-attribute inference.

---

> ### Author Response · Authors · 2026-04-25
> **Response to Reviewer UTcH: Clarity of Evidence**
>
> _**The changes related to this reviewer highlighted in blue in the revised manuscript.**_
>
> Thanks for acknowledging the timeliness of our work, broad experiments, and attention to governance and privacy risks.
>
> First, we address the concerns raised regarding the clarity of evidence supporting our claims.
>
> **1.1 Prompts ask models to “incorporate fairness” without specifying a formal notion.**
> To understand the impact of refined notions of fairness, we ran additional experiments with prompts including explicit fairness notions (exposure or representation). Using Approach 1 (sensitive attributes is specified in input), we refine the prompt accordingly and generate rankings. As seen below, explicit fairness specifications yielded only minor changes, supporting our simpler prompt design.
>
> | **Fairness instruction** | Initial Ranking | BASE *(Fairness without clarification)* | Fairness in representation | Fairness in exposure |
> |---|---|---|---|---|
> | **NDKL value** | 0.21 ± 0.07 | 0.06 ± 0.00 | 0.08 ± 0.04 | 0.10 ± 0.04 |
>
> In contrast, as our paper reports, the most substantial improvements in the metrics instead arise from the inclusion of few-shot examples. This explains why we focus our study on the latter.
>
> **1.2 Disentangling imitation from genuine reasoning.**
>
> As few-shot settings may induce imitation rather than deep reasoning, we soften the discussion accordingly (Sec. 6.4).  Further, to assess this, we compare few-shot example patterns with output rankings (M for male, F for female). Our analysis (below) indicates that LLMs are not strictly following (“blindly imitating”) the given examples (i.e., Male-Female pattern does not match), which instead appear to be serving as a mere guideline, as seen below.
>
> _Visual example of a shot and input-output rankings for 0, 1 and 2-shot scenarios produced using Gemini on Boston Marathon dataset. _:
>
> **Sample shot**
>
> __Input__ : MMMMMMMMMMMMFMFMMMFMMFFMFFFMFFFMFFFMFFFMFFFFFMFFMF
>
> __Output__: MMFFMFFMFFFMFFMFFMFFFMFFMFFMFFFMFFMFFMMMMMMMMMMMMM
>
> ---
> _Visual representation of generated test ranking (when varying shots):_
>
> __Setting: No shot__
>
> __I__ : MMMMMMMFMMMMMFFMMMFFMFMFFFMFFMFMFFFMFFMMFFFFFFFMFM
>
> __O__: MFMFMFMFMFMFMFMFMFMFMFMFMFMFMFMFMFMFMFMFMFMFMFMFMF
>
>
> __Setting: 1-shot__
>
> __I__ : MMMMMMMFMMMMMFFMMMFFMFMFFFMFFMFMFFFMFFMMFFFFFFFMFM
>
> __O__: MMFFMFFMFFFMFFFMFFMFFMFFFMFFFMFFFMFFMMMMMMMMMMMMMM
>
>
> __Setting: 2-shot__
>
> __I__ : MMMMMMMFMMMMMFFMMMFFMFMFFFMFFMFMFFFMFFMMFFFFFFFMFM
>
> __O__: MMFFMFFMFFMFFFMFFMFFMFFFMFFMFFMFFFMFFMMMMMMMMMMMMM
>
>
> **1.3**
> We agree and will note in the discussion that real-world group distributions are often imbalanced. We balanced our testing subsets deliberately as a controlled design choice to isolate fairness effects. But to explore your question, we added new experiments varying group proportions (Appendix A12). These results show that with lower disadvantaged-group proportions, LLMs continue to improve fairness.
>
> For “unknown” race labels, we follow prior work [1]. This removes uncertain cases, which we already discuss as a limitation in Section 5 and we expand upon it further in revision.
>
> Finally, as suggested by the reviewer, we soften our interpretation to reflect the specific scope of our methodology (in the Abstract and Sections 6 and 7).
>
> __Responses to Requested Changes__
>
> __C1. Fairness Specification:__ Addressed above in (1.1).
>
> __C2. Real-World Settings:__ Addressed above in (1.2).
>
> __C3. Statistical Significance:__ In the revision, we added stronger statistical analysis (Section 6.6). Beyond confidence intervals, we now use paired Wilcoxon signed-rank tests on key fairness (NDKL) and utility (NDCG@5) metrics across models, approaches, and shots. Main significant findings are summarized in the paper, full results are in the appendix, and modest differences are interpreted conservatively (Section 6.6). We thank the reviewer for this valuable suggestion.
>
> __C4.  Deployment Guidance:__ We added a new subsection on deployment guidance in Discussion section (Sec. 8), clarifying that implicit re-ranking without explicit sensitive attributes may still rely on latent attribute inference and therefore requires careful governance.
>
> __C5. Potential Misuse Pathways:__ We followed the suggestion to expand our discussion of potential impacts and misuse risks. In the revision in Sec. 8, we expand discussion of covert profiling, non-consensual sensitive-attribute inference, and fairness interventions that could bypass consent, transparency, or audit requirements. We also strengthen guidance on legal compliance, monitoring challenges, and safeguards for deployment.
>
> ---
> References
>
> [1] Ghosh A, et al. 2021. When Fair Ranking Meets Uncertain Inference. ACM SIGIR.
>
> [2] Ghosh A, et al. 2023. When fair classification meets noisy protected attributes. AIES.
>
> [3] Olulana O, et al. 2024. Hidden or Inferred: Fair Learning-to-Rank with Unknown Demographics. AIES.

---

### Review · Reviewer_s8Jq · 2026-04-13

**Summary Of Contributions:**

**Summary:**
This paper studies whether LLMs can do fairness-aware re-ranking when demographic attributes such as sex or race are missing, comparing three strategies: use third-party inferred attributes, let the LLM re-rank without explicit inference, or have the LLM explicitly infer attributes and then re-rank. Empirical results show that LLMs can sometimes improve group-fairness metrics while largely preserving utility, and that chain-of-thought prompting is the preferred strategy among their tested LLM pipelines. A second key finding is that explicit demographic inference is risky: when inferred attributes are noisy, fairness can degrade substantially, but stronger inference quality and few-shot demonstrations help. The paper positions this as an empirical audit of what LLM-based ranking pipelines do under demographic uncertainty.

**Strengths:**
- The paper tackles a real gap where most fair-ranking methods assume known protected attributes, while LLM ranking literature ignores fairness under missing demographics.
- The paper conducts an extensive comparison among external inference, implicit LLM behavior, and explicit LLM inference.
- The paper shows interesting findings: fairness is highly sensitive to attribute inference quality, and CoT prompting may bring the best gain.

**Weaknesses:**
- The technical contribution is quite limited. The evaluation framework is fairly simple, naive prompting with an external module/CoT/few-show samples.
- The presentation is not clear. The paper does not show the motivation for the three approaches and whether there are any other scenarios, i.e., finetuning. Fig 5-8 are not mentioned in the text, making them very confusing to interpret. Table 3 does not show the result of Approach 2.
- The experiments are too coarse-grained and general. In particular, the prompt deliberately treats fairness as a general concept. There are four datasets with two attributes (race & sex), but the analysis groups them together. I believe a fairness study should conduct a fine-grained analysis on each specific attribute. Including qualitative examples will also be very helpful.
- Minor typos: Appendix ?? on Page 9.

**Audience:**

Yes

**Audience Explanation:**

The paper studies the fairness of LLM ranking, which is related to TMLR audience.

**Claims And Evidence:**

No

**Claims Explanation:**

Partially supported. The experiments indeed show that fairness-aware re-ranking is sensitive, CoT helps, few-shot prompting helps. However, I find the definition of fairness in this work is too general. Although the paper deliberately designs the experiments as such, I believe it's not enough to conclude the fairness property of LLM ranking system.

**Requested Changes:**

- Improve the presentation and show the motivation for the framework/approach.
- Conduct deeper analysis on specific attributes along with qualitative examples.
- Provide ablation study of different components in Sec 4.1.2. For example, what if there is no Fairness Instruction, or if it is more fairness-specific.

---

> ### Author Response · Authors · 2026-04-25
> **Response to Reviewer s8jq: Clarifying Contribution, Strengthening Motivation, and Expanding Fairness Analysis**
>
> _**The changes related to this reviewer highlighted in orange in the revised manuscript.**_
>
> We thank the reviewer for pointing out the breadth of our experimental evaluation, the practical relevance of the gap we address, and the significance of our findings.
>
> __W1 – Clarifying Contribution__
>
> Our core contribution is not a new prompting mechanism but a systematic empirical investigation of fairness-aware LLM re-ranking under missing sensitive attributes. This is an important, underexplored setting given that such attributes are often withheld for legal and regulatory reasons. This aligns with prior work evaluating fairness-oriented tools and the broader challenges of assessing fairness interventions [1–3]. The value lies in our unified evaluation across multiple practical pipeline designs, models, datasets, and fairness metrics, yielding actionable insights on demographic uncertainty, implicit inference, and prompting design.
>
> __W2 - Improving Presentation Clarity and Experimental Motivation__
>
> __Motivate design choices.__ As suggested, we expand the design rationale in the revision (Sec 1, Our Approach & Scope of design choices considered), motivating the three approaches as practical deployment alternatives. We clarify that these are not exhaustive but represent the core pipeline design choices, consistent with prior literature on similar questions [1-3]. Our focus reflects realistic settings where organizations use existing LLM APIs in downstream decision pipelines, making task-specific fine-tuning often impractical. Optimizations such as fine-tuning are left for future work.
>
> __Refine presentation.__ We note that Figures 5–8 are referenced in Section 6.4 but discussed collectively. In the revision, we now address them individually, clearly mapping each observation to its corresponding chart and highlighting dataset/model differences and key trends. Regarding Table 3, the reviewer is correct that Approach 2 (Implicit) results were absent. Since it does not infer sex for candidates, inference accuracy cannot be computed. For clarity, we added a row for Approach 2 marked "Not Applicable."
>
> __W3 - Improving Fairness Specification and Analysis__
>
> While the paper examines fairness‑aware ranking behavior, our primary goal is to study such systems in the important setting where sensitive attributes are unavailable. We analyze whether the pipelines' performance generalizes across datasets and attributes, focusing on how different designs perform under demographic uncertainty rather than providing a comprehensive characterization of the fairness properties of LLM-based ranking systems.
>
> Yet, to explore the reviewer’s question regarding the fairness instruction, we conducted an additional experiment using extended prompt designs that incorporate two alternative fairness instructions: fairness in exposure and fairness in representation (see results in Table below Boston Marathon dataset using Approach 1).
>
> | **Fairness instruction** | Initial Ranking | BASE *(Fairness without clarification)* | Fairness in representation | Fairness in exposure |
> |---|---|---|---|---|
> | **NDKL value** | 0.21 ± 0.07 | 0.06 ± 0.00 | 0.08 ± 0.04 | 0.10 ± 0.04 |
>
> We find that these more specific notions lead to only minor differences relative to the baseline prompt for input data with sensitive attributes, whereas the most substantial improvements arise from the inclusion of few-shot examples. This supports why we work with the simpler prompt design to keep our study focused on our core research questions (see RQs in Introduction).
>
> __W4 – The latex compile error has now been addressed.__
>
> __Requested changes__
>
> __C1 – Improve Motivation & Presentation.__ In the new revision's introduction, we improve paragraph "State of the Art and limitations" to show motivation for the framework/approach (See Section 1 and in the Abstract).
>
> __C2 – Separate Analysis on Attributes.__ We indeed ran experiments on the sex and race attributes. We now add analysis comparing these in revision in Sections 6.1-6.6.
>
> __C3 – Ablation studies.__  The current manuscript already includes a targeted ablation of a central component of these approaches (Section 4.1.2), namely the few-shot examples, through comparisons between zero-shot and few-shot settings as well as sensitive-attribute-present (IMP-P) versus attribute-absent (IMP-A) demonstrations. This isolates the impact of two key components, namely, (a) exemplar guidance and (b) the role of explicitly conveying sensitive attributes through examples.
>
> Regarding the fairness instruction component, we now have added additional experiments that vary the fairness instructions (See Section A1 in Appendix).
>
> ---
> References
>
> [1] Ghosh, A. et al. 2021. When Fair Ranking Meets Uncertain Inference. SIGIR.
>
> [2] Ghosh, A. et al. 2023. When fair classification meets noisy protected attributes. AIES.
>
> [3] Olulana, O. et al. 2024. Hidden or Inferred: Fair Learning-to-Rank with Unknown Demographics. AIES.

---

### Decision · Action_Editor_FRPQ · 2026-05-22

**Recommendation:** Accept with minor revision

**Additional Comments:**

My main remaining concern is that the formal problem definition and the actual prompting implementation are still disconnected. This distinction matters because different LLMs may interpret the word “fairness” differently. As a result, the findings may not transfer straightforwardly across LLM backbones, prompt variants, sensitive attributes, or evaluation metrics. I acknowledge that the authors have already evaluated multiple LLMs and multiple fairness/utility metrics, and that the revision includes some additional prompt analysis. However, the main text should more explicitly state the limitation that the experiments audit LLM behavior under a particular generic fairness prompt, rather than demonstrating that LLMs optimize a well-defined or stable fairness objective.

**Audience:**

Yes

**Audience Explanation:**

The reviewers agree that this is a timely work to the LLM fairness community.

**Claims And Evidence:**

Yes

**Claims Explanation:**

The paper compares three pipeline designs (external demographic inference, implicit LLM-based re-ranking without explicit inference, and chain-of-thought LLM inference followed by re-ranking) across multiple datasets, LLM backbones, and inference services for LLM based fairness-aware reranking. The core contributions claimed in the introduction are backed by the empirical evidence.

---

> ### Author Response · Authors · 2026-05-31
> **Prompt-Objective Gap and Transferability of Findings**
>
> We thank the action editor for this observation.
>
> We have added a paragraph to Section 7 clarifying that the experiments audit LLM behavior under a deliberately underspecified fairness prompt, and that the observed improvements should be interpreted as empirical findings about these specific models under this prompt regime rather than as evidence of formal fairness optimization. We have also added a brief forward-reference in Section 3.2 to set reader expectations early. We believe this framing accurately reflects the scope of the contribution without undermining the empirical findings themselves.
>
> The changes are marked in green in the revision.